# A tachykinin-like neuroendocrine signalling axis couples central serotonin action and nutrient sensing with peripheral lipid metabolism

Lavinia Palamiuc[1,2], Tallie Noble[3], Emily Witham[2], Harkaranveer Ratanpal[2], Megan Vaughan[2,4] & Supriya Srinivasan[1,2]

Serotonin, a central neuromodulator with ancient ties to feeding and metabolism, is a major driver of body fat loss. However, mechanisms by which central serotonin action leads to fat loss remain unknown. Here, we report that the FLP-7 neuropeptide and its cognate receptor, NPR-22, function as the ligand-receptor pair that defines the neuroendocrine axis of serotonergic body fat loss in *Caenorhabditis elegans*. FLP-7 is secreted as a neuroendocrine peptide in proportion to fluctuations in neural serotonin circuit functions, and its release is regulated from secretory neurons via the nutrient sensor AMPK. FLP-7 acts via the NPR-22/Tachykinin2 receptor in the intestine and drives fat loss via the adipocyte triglyceride lipase ATGL-1. Importantly, this ligand-receptor pair does not alter other serotonin-dependent behaviours including food intake. For global modulators such as serotonin, the use of distinct neuroendocrine peptides for each output may be one means to achieve phenotypic selectivity.

[1] Department of Chemical Physiology and The Dorris Neuroscience Center, 1 Barnard Drive, Oceanside, California 92056, USA. [2] The Scripps Research Institute, 10550 North Torrey Pines Road, La Jolla, California, USA. [3] Mira Costa College, 1 Barnard Drive, Oceanside, California 92056, USA. [4] Kellogg School of Science and Technology, The Scripps Research Institute, 10550 North Torrey Pines Road, La Jolla, California, USA. Correspondence and requests for materials should be addressed to S.S. (email: supriya@scripps.edu).

The central nervous system plays a critical role in regulating energy balance and body fat stores, distinct from its effects on feeding behaviour. In recent years, strong evidence across metazoan systems has emerged to show that different brain regions potently control lipid metabolism in peripheral tissues, independently of changes in feeding behaviour[1–3]. Thus, discrete neural circuits and neuronal subpopulations regulate body fat metabolism. Although for certain central regulators the neural circuits governing peripheral metabolism are now beginning to be understood, identifying neuroendocrine factors that selectively govern fat metabolism has remained a long-standing challenge. Additionally, the extent to which nutrient sensing by the brain modulates peripheral metabolism via such neuroendocrine factors is poorly understood.

The ancient central nervous system neuromodulator serotonin (5-hydroxytryptamine, 5-HT) is a valuable paradigm for the study of global regulators of animal physiology. In metazoans, 5-HT controls food intake and feeding behaviour, mood, adiposity, locomotion and energy expenditure[4–6]. In humans, with respect to the control of energy balance, global serotonergic agonists and antagonists have long been administered to suppress appetite, increase energy expenditure, or both[7,8]. However the limited efficacy of these global serotonin-boosting compounds is often accompanied by substantial side effects, and has mitigated their use. Yet, the importance of 5-HT signalling in stimulating fat metabolism across many species suggests that the identification of selective factors that function downstream of central 5-HT to stimulate fat metabolism would be valuable. A body of evidence has revealed a broad framework for the effects of neuronal 5-HT on whole body energy balance by exploiting the tractability of the *C. elegans* system[9,10]. Manipulations of 5-HT levels via exogenous administration or by endogenous control using genetic approaches reveal that, as in mammals, neuronal 5-HT is a potent stimulator of body fat loss and energy expenditure[11].

In the nematode *C. elegans*, 5-HT is synthesized by the conserved rate-limiting enzyme tryptophan hydroxylase (TPH-1) in only a few pairs of neurons, and is not present in the intestine or in other metabolic tissues under normal conditions[5]. Loss of the TPH-1 enzyme leads to undetectable levels of neuronal 5-HT and a significant increase in body fat reserves accompanied by decreased energy expenditure. Functional studies reveal that of all the biogenic amine receptors in the genome, a 5-HT-gated chloride channel called MOD-1 is the sole receptor essential for body fat loss[1]. In the intestine, the major metabolic organ for *C. elegans* and the predominant site for fat metabolism, the rate-limiting enzyme adipocyte triglyceride lipase ATGL-1 is transcriptionally induced by 5-HT signalling and serves to stimulate fat breakdown via hydrolysis of stored triglycerides to fatty acids[1]. RNAi-based screens also revealed that increased 5-HT signalling elicits a cascade of β-oxidation enzymes in the intestine that convert fatty acids to energy in the mitochondria[11].

Despite its neuronal origins, the metabolic effects of 5-HT occur in the intestine. In our previous work, we deciphered the neural circuit for 5-HTergic fat mobilization in *C. elegans*[1]. We found that 5-HT synthesis is required in the ADF chemosensory neurons whereas MOD-1 the critical serotonergic channel is necessary and sufficient in a single pair of neurons called URX, which receive direct synaptic input from the ADF neurons. Additionally, we found that octopamine (OA), the invertebrate analogue of adrenaline, provides a permissive cue to maintain 5-HT signalling via regulating *tph-1* expression and 5-HT levels in the ADF neurons. Thus, the primary components of the 5-HT pathway: the biosynthetic enzyme and the receptor, reside in the nervous system, and the collective evidence indicates that 5-HTergic regulation of body fat loss occurs indirectly, perhaps via the relay of neuroendocrine factor/s from the nervous system, to the intestine[1]. It has remained a challenge to identify selective neuroendocrine factors that stimulate body fat mobilization downstream of central mediators of energy balance, in any system. Pioneering biochemical approaches were used to identify neuropeptide hormones that communicate in endocrine fashion from the mammalian hypothalamus to control the physiology of stature, reproduction and other aspects of whole animal physiology[12–14]. Despite these immense advances, biochemical approaches relying on the relative abundance of peptides in the mammalian hypothalamus did not lead to the identification of neuroendocrine hormones that control body weight, and endocrine factors that potently stimulate body fat loss have since remained unknown.

In this study, we identify a secreted neuropeptide ligand and its cognate receptor that constitute the core 5-HT neuroendocrine axis and selectively stimulates body fat loss in *C. elegans*. The ligand is secreted in proportion to 5-HT circuit functions and the activity of the nutrient sensor AMPK, in the secretory neurons. In the intestine, activation of the receptor promotes fat loss via induction of the ATGL-1 lipase. The broad conservation of this signalling axis suggests that such approaches are valuable in identifying novel and selective neuroendocrine factors that underlie the central control of body fat metabolism.

## Results

**Neuropeptide signalling is required for 5-HT-mediated fat loss.** Our previous work describing the 5-HTergic neural circuit revealed that rather than 5-HT itself, an unknown neuroendocrine factor is released from the nervous system and relayed to metabolic tissues to stimulate body fat loss[1]. The *C. elegans* intestine is not directly innervated (www.wormatlas.org) and therefore offers a valuable platform to identify neuroendocrine factors that communicate between the nervous system and the metabolic tissues. The diversity of known mechanisms of neuroendocrine signalling across different species prompted us to use a process of elimination followed by a screen. To begin investigating the nature of this neuroendocrine signal, we first measured the extent to which serotonin-mediated fat loss was dependent upon the release of canonical neurotransmitters (acetylcholine, γ-amino butyric acid and glutamate), versus that of neuropeptidergic signals. In the nervous system, canonical neurotransmitters are localized to clear synaptic vesicles, which require a protein called UNC-13 (MUNC-13 in mammals) for fusion with the plasma membrane at the synapse[15,16]. On the other hand, neuropeptides and biogenic amine neurotransmitters are localized to dense core vesicles, which require the conserved calcium-dependent activator protein (CAPS) or UNC-31/CAPS in *C. elegans*[17–19]. Both *unc-13* and *unc-31* are broadly expressed in the *C. elegans* nervous system, and not in other tissues. Thus, loss of unc-13 function leads to a block in the release of the canonical neurotransmitters, whereas loss of unc-31 blocks the release of neuropeptides and biogenic amines. We measured the extent to which *unc-13* and *unc-31* mutants were essential in promoting 5-HT-mediated fat loss. With respect to body fat content, vehicle-treated *unc-13* mutants resembled wild-type animals; however, *unc-31* mutants had ∼50% greater body fat than either genotype (Fig. 1a), suggesting that the contents of dense core vesicles from the nervous system regulate fat stores under basal conditions. As reported previously[1,11], 5HT-treated wild-type animals retained approximately 40–50% of the body fat seen in vehicle-treated controls, as did the *unc-13* mutants (Fig. 1a). On the other hand, 5-HT-treated *unc-31* mutants fully suppressed serotonergic

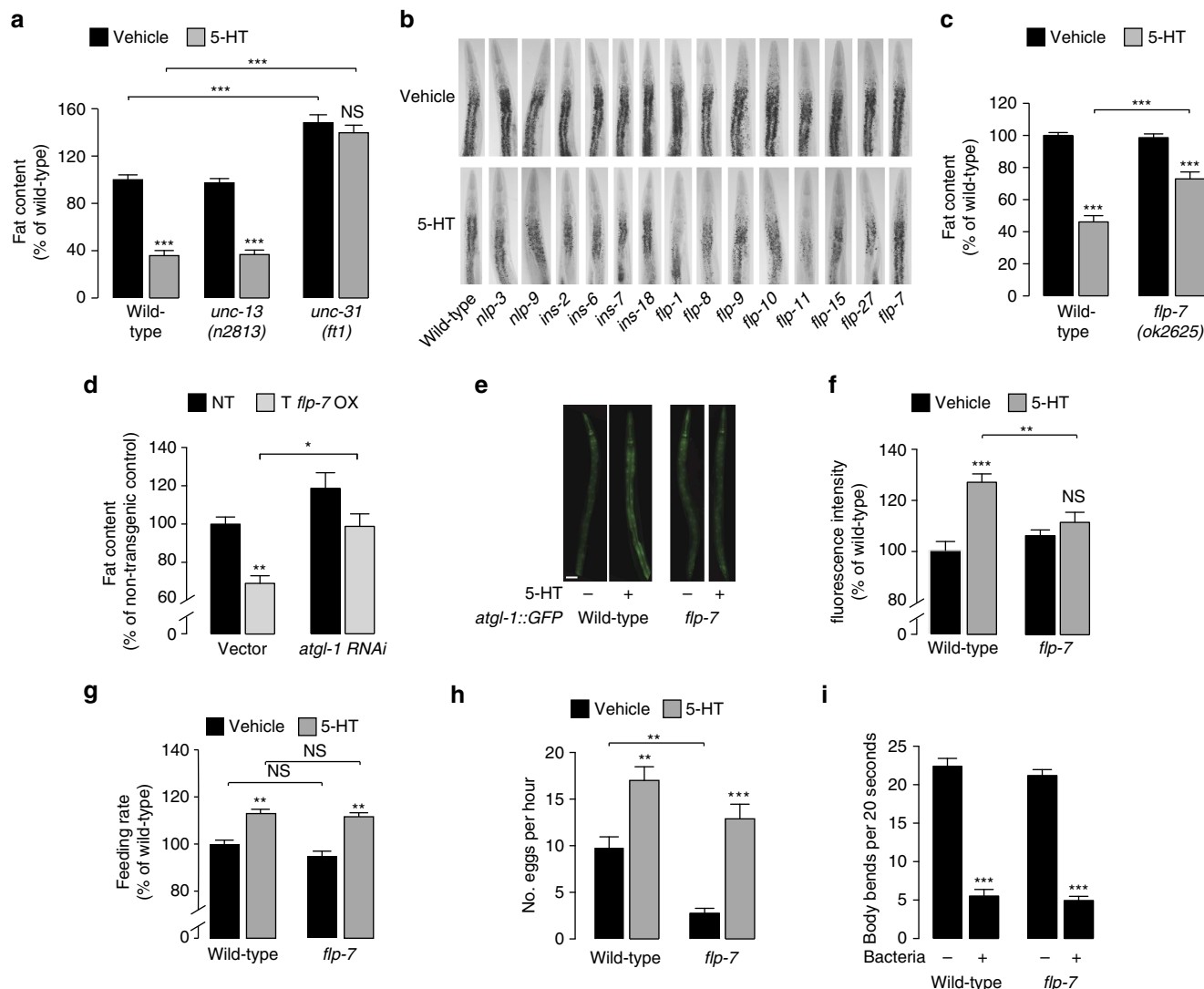

**Figure 1 | Neuropeptide signalling is required for 5-HT-mediated fat loss.** (**a,c**) Vehicle- and 5-HT-treated animals were fixed and stained with oil Red O. Genotypes are indicated in the figure. Fat content for each genotype was quantified and expressed as a percentage of vehicle-treated wild-type animals ± s.e.m. ($n = 10$–14). ***$P < 0.001$ and NS, not significant by two-way ANOVA. (**b**) Representative images of vehicle- or 5-HT-treatment, fixed and stained with oil Red O in the indicated genotypes. Animals are oriented facing upwards with the pharynx at the anterior end. Lipid droplets that store body fat are restricted to the intestinal cells. (**d**) Wild-type animals (black bars) overexpressing the *flp-7* transgene (OX; grey bars) grown on vector or *atgl-1 RNAi* containing bacteria were fixed and stained with oil Red O. Data are expressed as a percentage of body fat in wild-type (non-transgenic) animals ± s.e.m. (lower panels; $n = 11$–21). *$P < 0.05$ and **$P < 0.01$ by two-way ANOVA. (**e**) Representative images of vehicle- and 5-HT-treated wild-type animals and *flp-7* mutants bearing an integrated *atgl-1::GFP* transgene. Scale bar, 50 μm. (**f**) The fluorescence intensity of *atgl-1* expression in vehicle- and 5-HT-treated wild-type animals and *flp-7* mutants was quantified and is expressed as a percentage of vehicle-treated wild-type animals ± s.e.m. (17–27). **$P < 0.01$, ***$P < 0.001$ and NS, not significant by two-way ANOVA. (**g**) Feeding rate is expressed as a percentage of vehicle-treated wild-type animals ± s.e.m. ($n = 27$). Genotypes are indicated in the figure **$P < 0.01$ and NS, not significant by two-way ANOVA. (**h**) Egg-laying rates were measured in vehicle- and 5-HT-treated wild-type animals and *flp-7* mutants. For each genotype and condition, the average number of eggs laid was counted as described in the methods. Data are expressed as an average ± s.e.m. ($n = 10$). **$P < 0.01$ and ***$P < 0.001$, by two-way ANOVA. (**i**) The number of body bends over a 20-s interval was counted in the presence and absence of the bacterial food so'urce. Data are expressed as an average ± s.e.m. ($n = 10$–16). ***$P < 0.001$ by two-way ANOVA.

fat loss and retained as much body fat as the vehicle-treated controls (Fig. 1a). Thus, a UNC-31/CAPS-dependent secretory process is required for the effects of 5-HT on body fat loss.

The *C. elegans* genome encodes 113 neuropeptide genes, and biosynthetic enzymes for four biogenic amines: 5-HT itself, octopamine (OA), dopamine and tyramine[20,21]. Our previous work had delineated a role for neuronal OA, the invertebrate analogue of noradrenaline, in providing a permissive cue to maintain 5-HT synthesis in the ADF chemosensory neurons. Loss of the dopamine and tyramine biosynthetic enzymes did not

lead to appreciable differences in body fat with 5-HT treatment[1]. In addition, we reported that 5-HT-induced fat loss was independent of the *daf-2* insulin/IGF receptor and its downstream partner, *daf-16*/FOXO (ref. 11). Therefore we next assessed the *C. elegans* neuropeptide genes for roles in 5-HT-mediated fat loss. Two predominant neuropeptide gene families in *C. elegans* include the *flp* and *nlp* genes[22], many of which are functional orthologues of vertebrate neuropeptides including oxytocin, vasopressin and the neuromedin family[23–25]. An additional neuropeptide family called the *ins* genes bear

sequence homology to the insulin peptide family[26]. To identify neuropeptides that relay the effects of neuronal 5-HT, we screened the existing mutants of the *flp/nlp/ins* neuropeptide gene families (77/113 genes; Fig. 1b) for suppression of 5-HT-induced fat loss. One neuropeptide gene called *flp-7* emerged as a robust partial suppressor (Fig. 1b,c and Supplementary Fig. 1a). Relative to wild-type animals, 5-HT-treated *flp-7(ok2625)* null mutants retained a significantly greater proportion of body fat (Fig. 1c; ∼45% versus ∼75%), whereas vehicle-treated *flp-7* mutants resembled wild-type animals. These data suggested that *flp-7* is required for 5-HT-induced fat loss. In wild-type animals, constitutive overexpression of *flp-7* under its own promoter led to a readily apparent decrease in body fat reserves (Fig. 1d), suggesting that its activity promotes body fat loss.

**Physiological properties of FLP-7.** In examining the metabolic effects of 5-HT in the intestine, we had previously uncovered a role for ATGL-1, a conserved rate-limiting enzyme that converts triglycerides into fatty acids[27] that are in turn, utilized for β-oxidation and energy production in the mitochondria[1,11]. A reporter line and qPCR experiments had shown that 5-HT treatment induces transcription of *atgl-1* (ref. 1). Thus, we examined whether *flp-7* was required for 5-HT-mediated *atgl-1* induction. *flp-7* mutants show a near-complete suppression of the transcriptional induction of the *atgl-1::GFP* reporter line in 5-HT-treated animals (Fig. 1e,f). In addition, RNAi-mediated inactivation of *atgl-1* abrogated the increased fat loss seen with *flp-7* overexpression (Fig. 1d). These experiments show that the effects of 5-HT on atgl-1-mediated fat loss require flp-7 signalling.

In *C. elegans*, in addition to its effects on body fat mobilization, neuronal 5-HT regulates food intake[5], reproduction and locomotor changes based on food availability[6,28]. Yet, the 5-HTergic regulation of body fat stores employs a distinct neuronal circuit and regulatory cascade in the intestine[1,11]. To address the question of how such selectivity might be achieved, we measured the extent to which *flp-7* is required for other phenotypes associated with the food-5-HT signalling axis. We found that loss of the *flp-7* gene did not lead to changes in 5-HT-induced food intake (Fig. 1g), reproduction (Fig. 1h) or the enhanced slowing response, a satiety-like locomotor response to food availability (Fig. 1i). We note that *flp-7* mutants have a decreased egg-laying rate relative to wild-type animals (Fig. 1h); however, the decreased egg-laying did not correlate with changes in body fat because vehicle-treated *flp-7* mutants do not display a body fat phenotype. Instead, *flp-7* mutants suppress 5-HT-induced fat loss without suppressing 5-HT-stimulated egg-laying. Thus the serotonergic effects on egg-laying and body fat loss are not causally linked by flp-7. Thus, under conditions of serotonergic stimulation, the FLP-7 neuropeptide selectively stimulates body fat loss. The *flp-7* gene encodes peptides that resemble the mammalian tachykinin peptide family exemplified by Substance P (Supplementary Fig. 1b,c) and are secreted neuropeptides regulating gut motility[29], but have not previously been associated with fat metabolism. Our data suggest that FLP-7 is essential for serotonin-dependent fat loss, and that its overexpression leads to a substantial reduction in body fat stores.

**The ASI neurosecretory neurons are the site of FLP-7 synthesis.** A fluorescent transgenic reporter line revealed that *flp-7* is expressed in a few pairs of head neurons (Fig. 2a). We wanted to determine the identity of the neurons from which FLP-7 expression regulates 5-HT-induced body fat loss. A *flp-7*-rescuing

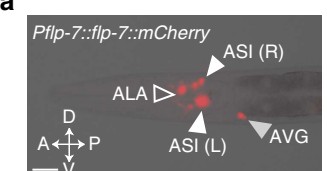

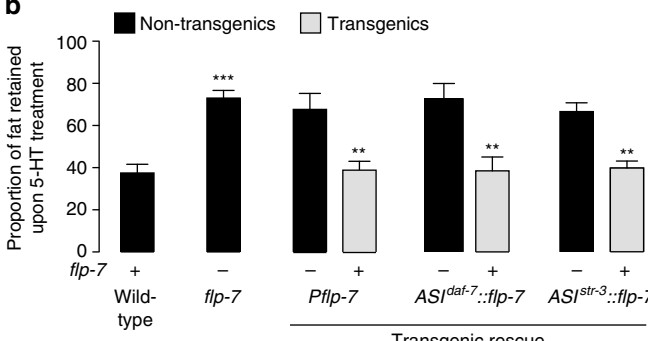

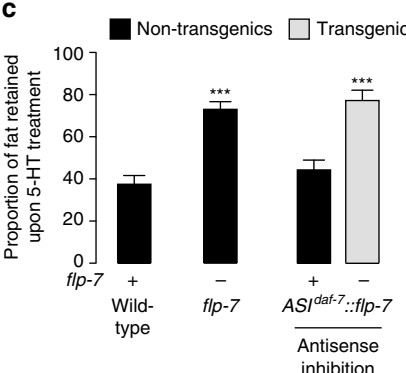

**Figure 2 | The tachykinin-related neuropeptide FLP-7 functions in the ASI neurons to regulate 5-HT-induced fat loss.** (**a**) Fluorescent image of a transgenic animal bearing a polycistronic *flp-7::mCherry* transgene under the control of the endogenous *flp-7* promoter. White arrowheads indicate expression in ASI neurons, the open arrowhead indicates expression in ALA and the grey arrowhead indicates expression in AVG. A, anterior; P, posterior; V, ventral; D, dorsal. Scale bar, 20 μm. (**b**) Fat content of vehicle- and 5-HT-treated *flp-7* mutants bearing a *flp-7* transgene using the indicated promoters was measured. Relative to non-transgenic controls, transgenic *flp-7* animals bearing the *flp-7* transgene under the control of the endogenous *flp-7* promoter and the heterologous ASI promoters *daf-7* and *str-3* restore 5-HT-induced fat loss indistinguishably from wild-type animals. Data are expressed as a proportion of fat retained upon 5-HT treatment ± s.e.m. (lower panels; $n = 8$–20). **$P < 0.01$ and ***$P < 0.001$ by two-way ANOVA. (**c**) *flp-7* was inactivated in wild-type animals using RNAi-mediated antisense. Relative to non-transgenic controls, transgenic wild-type animals bearing *flp-7* sense-antisense transgenes under the control of the ASI-specific *daf-7* promoter suppress 5-HT-induced fat loss, as seen in *flp-7* mutants. Data are expressed as a proportion of fat retained upon 5-HT treatment ± s.e.m. (lower panels; $n = 10$–24). ***$P < 0.001$ by two-way ANOVA.

transgene under the control of its own promoter showed vivid expression in a few pairs of head neurons (but not in the intestine, the muscles or the hypodermis) and fully restored 5-HT-induced fat loss in the intestine (Fig. 2a,b). We observed reproducible *flp-7* expression in the ALA and the AVG interneurons and in the ASI sensory neurons (Supplementary Fig. 2). The ALA motor neuron is known to regulate locomotion[30]

and the AVG neuron is essential for ventral cord development[31]. On the other hand, the ASI sensory neuron pair regulates whole body physiology during development and is involved in lifespan control via the secretion of regulatory peptides[32–35]. We tested the sufficiency of *flp-7* expression in ASI neurons using two independent promoters, *daf-7* and *str-3*, both of which fully restored 5-HT-induced fat loss observed in *flp-7* mutants (Fig. 2b), to the same extent as seen in wild-type and *flp-7* transgenic animals. Additionally, RNAi-mediated anti-sense inhibition of the *flp-7* gene in the ASI neurons recapitulated the *flp-7* mutant phenotype and led to robust suppression of 5-HT-induced fat loss (Fig. 2c). Thus, *flp-7* expression in ASI neurons is necessary and sufficient to maintain 5-HTergic fat loss.

**Visualizing FLP-7 neuropeptide secretion in living animals.** Although ASI neurons are the source of FLP-7, its effects on fat loss occur in the intestine, the major site for body fat storage and lipid metabolism[9]. Thus, we hypothesized that FLP-7 is released as a secreted peptide from the ASI neurons. Previous studies have shown that secreted neuropeptides tagged with genetically encoded fluorescent markers can be taken up by coelomocytes, scavenger cells that non-specifically endocytose and thus concentrate the contents of the coelomic fluid, which functions as the circulatory system for *C. elegans*[36,37]. Thus, the coelomocyte uptake assay can be used to evaluate the extent to which a neuropeptide of interest is found in the circulation (Fig. 3a). We simultaneously expressed and integrated two transgenes in wild-type animals: one bearing the *flp-7* gene fused to mCherry and expressed from the ASI neurons (*ASI*[daf-7]*::flp-7mCherry*), and the other bearing a GFP-based marker expressed solely in the coelomocytes (CLM::GFP). We found that the FLP-7mCherry fusion protein expressed in the ASI neurons accumulates in the coelomocytes of wild-type animals. Exogenous 5-HT treatment induced a robust increase in FLP-7 secretion (~160% of vehicle-treated controls; Fig. 3b–d). Consistent with the dependence of 5-HT-induced fat loss on dense core vesicle function (Fig. 1a), *unc-31* mutants showed a profound decrease in FLP-7mCherry accumulation in the coelomocytes, and subjecting these mutants to 5-HT treatment had no discernible effect on FLP-7mCherry secretion (Fig. 3b–d). Thus, FLP-7mCherry accumulation in the coelomocytes is dependent on dense core vesicle release from neurons. We incidentally observed an accumulation of un-secreted FLP-7mCherry protein in the ASI neurons of *unc-31* mutants (not shown).

Several measures were employed to ensure that the observed coelomocyte uptake of FLP-7mCherry was a true consequence of FLP-7 secretion. First, we observed no changes in the expression levels of the *flp-7mCherry* mRNA following 5-HT treatment, or in the mutants in which endogenous 5-HT levels are altered: *tph-1* (no 5-HT synthesis) and *mod-5* (no 5-HT reuptake; increased synaptic 5-HT; Supplementary Fig. 3a). Because we use the *daf-7* ASI promoter to express the *flp-7mCherry* transgene, we also measured endogenous *daf-7* mRNA levels. Although *tph-1* mutants did have decreased *daf-7* mRNA, *mod-5* mutants and exogenous 5-HT treatment did not (Supplementary Fig. 3b). A *daf-7::GFP* reporter line also did not show changes in GFP fluorescence upon 5-HT treatment (Supplementary Fig. 3c). The decrease in *daf-7* mRNA in *tph-1* mutants was not reflected in the expression level of the *flp-7mCherry* transgene, which remained unaltered (Supplementary Fig. 3a). It is possible that *daf-7* mRNA regulation by *tph-1* occurs via post-transcriptional effects. Finally, FLP-7 fluorescence intensity in the ASI neurons also did not change in any of the genotypes or

conditions tested (Supplementary Fig. 3d,e). Thus, the observed changes in FLP-7mCherry punctae in the coelomocytes cannot be attributed to differences in *flp-7* expression in the ASI neurons.

To independently verify the secretion of FLP-7mCherry from the ASI neurons we used a different promoter called *str-3*, which is not regulated by *tph-1* or 5-HT (refs 38,39). An *ASI*[str-3]*::flp-7mCherry* transgenic line was generated, and upon 5-HT-treatment we again observed robust secretion of FLP-7mCherry and accumulation in the coelomocytes (Supplementary Fig. 3f,g). Second, across all experimental conditions tested (described below), there were no appreciable differences in GFP expression intensity in the coelomocytes (Fig. 3e). Finally, for all experimental animals and groups, a correlation plot of FLP-7mCherry and GFP expression intensity revealed no observable trend (correlation coefficient 0.15; Fig. 3f). Thus, mCherry and GFP expression intensity are independent of one another, and the ratio of FLP-7mCherry punctae to coelomocyte GFP expression can be used as a reliable indicator of FLP-7 secretion under different experimental conditions.

We wanted to assess whether FLP-7 secretion is modulated by changes in endogenous 5-HT levels in neurons. We crossed the transgenic FLP-7mCherry secretion lines into *tph-1* (tryptophan hydroxylase null) and *mod-5* (5-HT reuptake transporter null) mutant animals, which result in no 5-HT synthesis and in increased synaptic 5-HT, respectively. *tph-1* and *mod-5* are each expressed solely in a few pairs of head neurons and not in the intestine or in other metabolic tissues. As judged by the accumulation of FLP-7mCherry punctae in the coelomocytes, *tph-1* mutants showed a significant decrease in FLP-7 secretion (~60% of wild-type), whereas *mod-5* mutants had substantially increased FLP-7 secretion (~135% of wild-type; Fig. 3b–d and Supplementary Fig. 3f,g). The alterations in FLP-7 secretion in both mutants correlate with their corresponding body fat phenotypes: *tph-1* mutants have increased body fat, and *mod-5* mutants show a reduction in fat content[1,11]. Additionally, as described above, overexpression of *flp-7* led to decreased body fat (Fig. 1d). Together, these experiments show that FLP-7 secretion from the ASI neurons is responsive to changes in neuronal 5-HT signalling, and that it drives fat loss in the intestine.

**Circuitry of 5-HT- and OA-induced FLP-7 secretion.** We had previously shown that the serotonergic MOD-1 chloride channel and the octopaminergic SER-6 GPCR function in an integrated neuronal circuit to regulate fat loss via an endocrine signal. To determine the extent to which each of these genes alters FLP-7 secretion properties, we crossed the respective null mutants into the FLP-7mCherry secretion line. Loss of either *mod-1* or *ser-6* did not appreciably decrease FLP-7mCherry secretion under basal conditions in vehicle-treated animals. However, both single mutants suppressed the 5-HT-induced secretion of FLP-7mCherry (Fig. 4a,b,d), in keeping with their previously observed effects on suppression of serotonergic fat loss[1]. The suppression of FLP-7mCherry secretion in the *mod-1;ser-6* double mutant is also consistent with the previously observed complete suppression of 5-HT-induced fat loss. To corroborate the effects of *mod-1* and *ser-6* on FLP-7mCherry secretion with changes in body fat content, we generated double mutants between each of the receptor mutants and the *flp-7* peptide mutant. We found that the *mod-1;flp-7* and *ser-6;flp-7* mutants did not show appreciable differences in body fat content upon vehicle treatment alone (Supplementary Fig. 4), and both double mutants showed robust suppression of 5-HT-mediated fat loss, as did the *mod-1;ser-6;flp-7* triple mutant (Supplementary Fig. 4). Together, our results suggest that mod-1 and ser-6 are

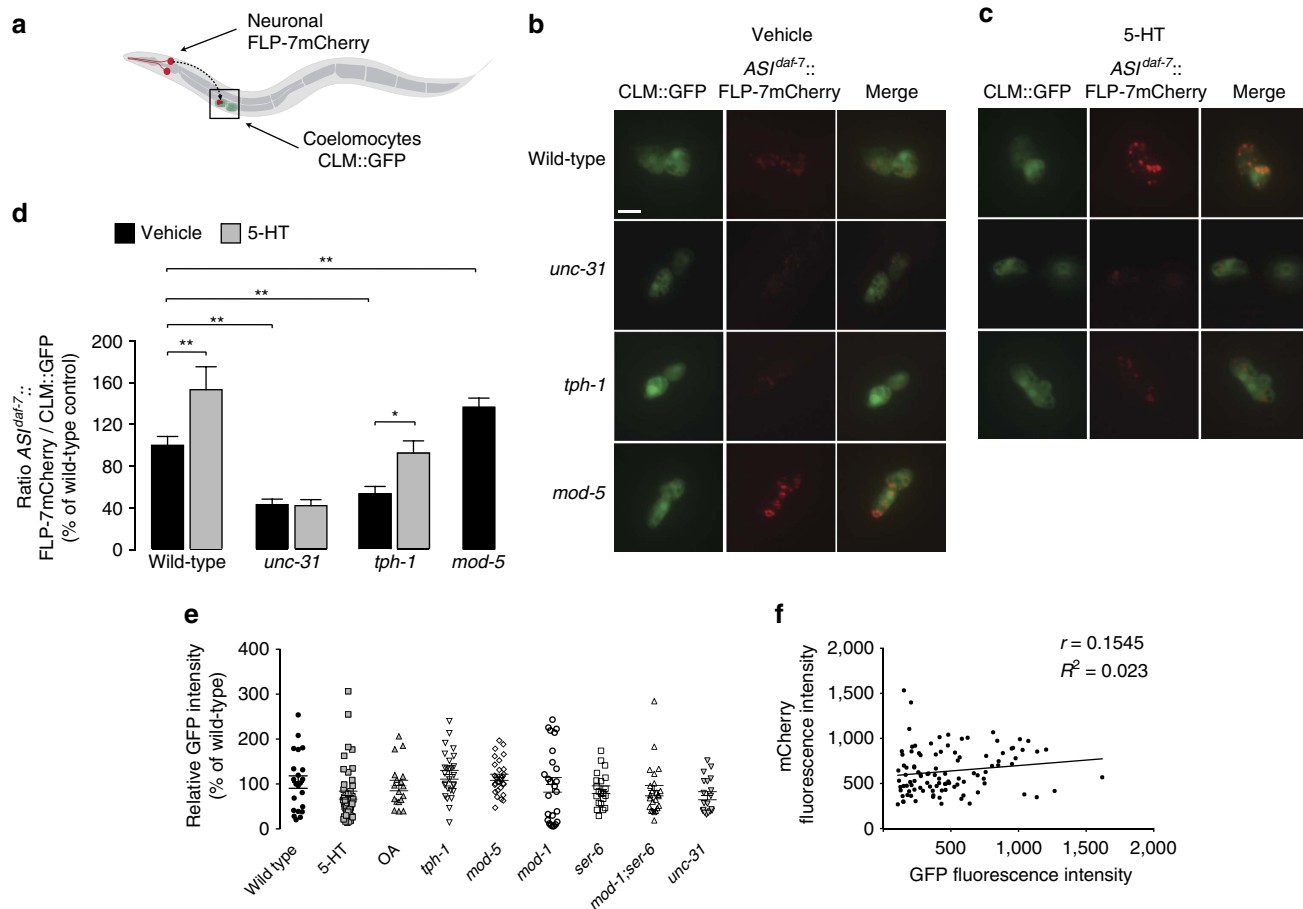

**Figure 3 | The coelomocyte uptake assay allows visualization of FLP-7 secretion in response to serotonergic genes.** (**a**) Model illustrating the coelomocyte uptake assay for neuropeptide secretion. The FLP-7mCherry fusion protein (marked in red) is expressed from ASI neurons and GFP is expressed in the coelomocytes (marked in green). The ratio of red:green fluorescence is used to quantify the extent of secretion under different experimental conditions. CLM, coelomocytes. (**b,c**) Representative images of vehicle- and 5-HT-treated wild-type, *unc-31*, *tph-1* and *mod-5* animals bearing the FLP-7mCherry and CLM::GFP integrated transgenes, respectively. Left panels, GFP expression in coelomocytes; centre panels, secreted FLP-7mCherry uptake in coelomocytes; right panels (merge). Scale bar, 10 μm. (**d**) For vehicle- and 5-HT-treated animals bearing integrated FLP-7mCherry and CLM::GFP transgenes, the intensity of FLP-7mCherry fluorescence within a single coelomocyte was quantified and normalized to the area of CLM::GFP expression. Genotypes are indicated in the figure. Data are expressed as a percentage of the normalized FLP-7mCherry fluorescence intensity of vehicle-treated wild-type animals ± s.e.m. (n = 10–20 animals). *P < 0.05 and **P < 0.01 by two-way ANOVA. (**e**) Individual values for the fluorescence intensity of CLM::GFP within a single coelomocyte are shown for each condition. Bars indicate the average value ± s.e.m. within each condition. Data are expressed as a percentage of wild-type animals. No significant differences were observed by one-way ANOVA, n = 19–46. (**f**) mCherry fluorescence intensity values are plotted against GFP fluorescence intensity values for each animal across representative experimental conditions, n = 103.

both required for the 5-HT-dependent secretion of FLP-7 and the ensuing effects on body fat loss.

In previous work we showed that octopamine (OA), the invertebrate analogue of noradrenaline, stimulates fat loss via a circuit integrated with 5-HT signalling. OA-induced fat loss functions via the SER-6 octopaminergic receptor in the AWB neurons, which in turn maintains *tph-1* expression in ADF neurons[1]. In addition, *ser-6* and *tph-1* mutants suppress the induction of fat loss via exogenous octopamine. We therefore tested the effects of exogenous OA on FLP-7 secretion in wild-type, *ser-6* and *tph-1* animals. Interestingly, *flp-7* mutants suppress OA-induced fat loss (Fig. 5a). Exogenous OA treatment increased FLP-7mCherry secretion from the ASI neurons (Fig. 4a,c,d, 5b–d and Supplementary Fig. 5), and *unc-31* mutants in which dense core vesicle secretion is impaired, suppress FLP-7mCherry secretion both under basal and OA-stimulated conditions (Fig. 5b–d). In addition, as predicted, *tph-1* mutants also showed a decrease in OA-induced FLP-7mCherry secretion (Fig. 5b–d). *ser-6*, *mod-1* and

*mod-1;ser-6* mutants suppress OA-mediated FLP-7mCherry secretion to approximately the same extent (Fig. 4a,c,d). Thus, as seen with 5-HT, the effects of OA on FLP-7mCherry secretion occur via the integrated effects of SER-6 and MOD-1 signalling. As described above, the changes in the accumulation and intensity of FLP-7mCherry in the coelomocytes were independent of GFP expression intensity in the coelomocytes, which did not change under these conditions (Fig. 3e,f). Together with our previous studies describing the site of action of MOD-1 and SER-6, the localization of FLP-7 to the ASI neurons suggests a circuit in which 5-HT signalling via the ADF and URX sensory neurons is integrated with OA signalling via the RIC and AWB neurons to modulate the extent of FLP-7 secretion from the ASI neurons.

**AMPK signalling in the ASI neurons regulates FLP-7 secretion.** We wanted to investigate the mechanism by which the ASI neurons regulate FLP-7 release. In independent experiments,

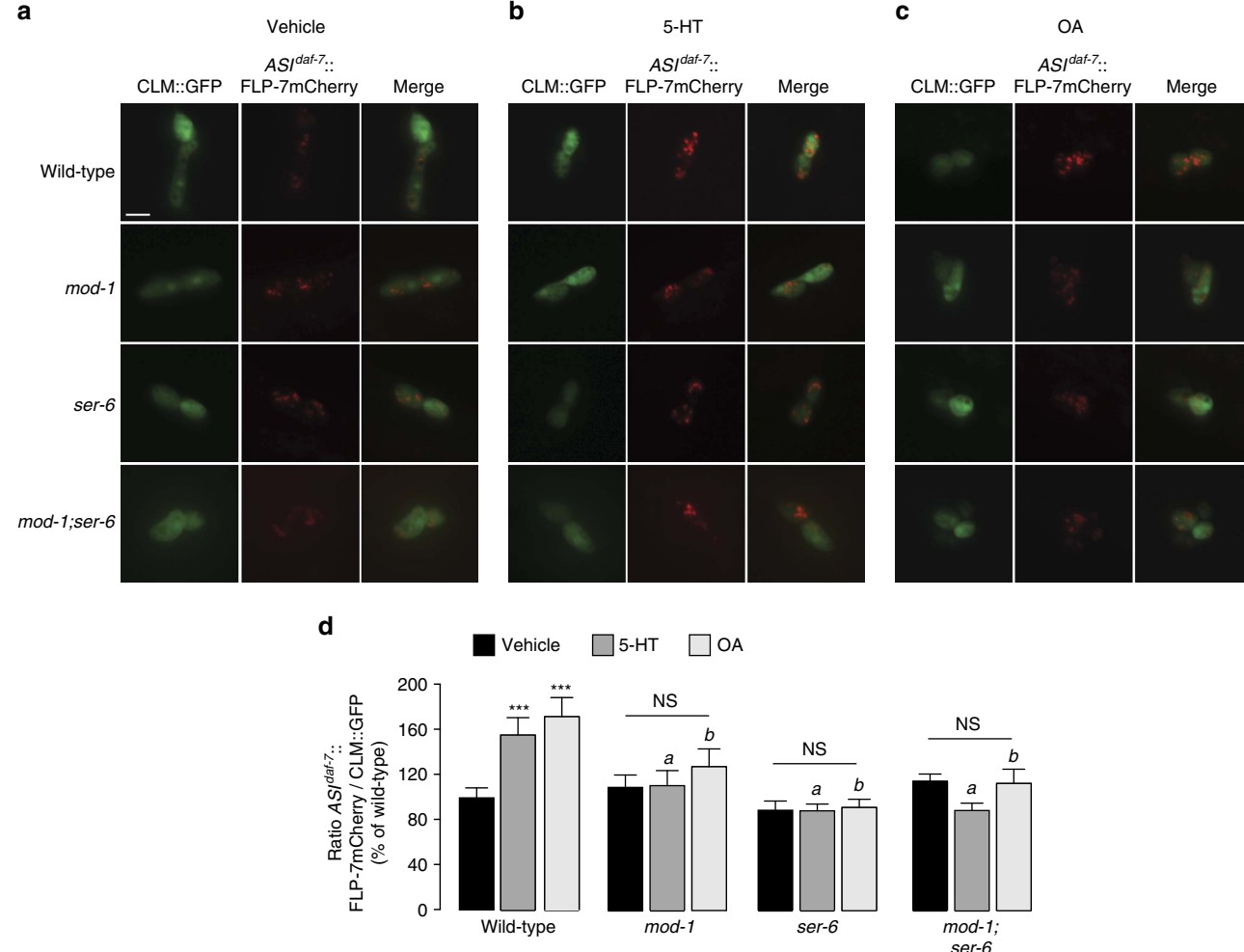

**Figure 4 | FLP-7 secretion in response to 5-HT and OA receptors.** (a–c) Representative images of vehicle-, 5-HT- and Octopamine (OA)-treated wild-type, *mod-1*, *ser-6*, and *mod-1;ser-6* animals bearing integrated FLP-7mCherry and CLM::GFP transgenes, respectively. Left panels, GFP expression in coelomocytes; centre panels, secreted FLP-7mCherry uptake in coelomocytes; right panels (merge). Scale bar, 10 μm. (d) The intensity of FLP-7mCherry fluorescence was quantified and normalized to the area of the CLM::GFP. Genotypes are indicated in the figure. Data are expressed as a percentage of the normalized FLP-7mCherry fluorescence intensity of vehicle-treated wild-type animals ± s.e.m. (lower panels; *n* = 11–27). ***P < 0.001 and NS, not significant by two-way ANOVA for within-group analyses. (a) P < 0.01 by two-way ANOVA compared to 5-HT-treated wild-type animals. (b) P < 0.05 by two-way ANOVA compared to OA-treated wild-type animals.

we had observed that loss of *aak-2*, the *C. elegans* orthologue of the AMP kinase α subunit (AMPK) led to a reduction in body fat stores, to approximately the same extent as 5-HT treatment. AMPK is an ancient nutrient sensor that is activated by low cellular energy states (increased AMP:ATP ratio) and by hormones[40]. 5-HT is well known to be an indicator of food presence and increased nutrient availability in *C. elegans*[9]; therefore, it was plausible to consider a role for AMPK in transducing 5-HT-encoded nutrient information. Evidence from the literature also suggested that loss of *aak-2* mimics increased 5-HT signalling with respect to feeding behaviour[41]. To evaluate the possibility that AMPK signalling regulates 5-HT-mediated FLP-7 secretion, we crossed the *flp-7mCherry* secretion line with the *aak-2* mutants, and observed a robust increase in FLP-7mCherry secretion (Fig. 6a–c). This increase in FLP-7 secretion was accompanied by a ∼60% reduction in body fat stores in the *aak-2* mutants (Fig. 6d). *aak-2* cDNA under an ASI-specific promoter in these mutants significantly restored body fat levels (Fig. 6d); thus, AAK-2/AMPK signalling from ASI neurons regulates body fat stores.

Recent studies have identified a transcription factor called CRTC-1, a co-regulator of the transcription factor CREB (cAMP response element binding protein), as one target of AMPK with respect to its role in lifespan control[42,43]. Neuronal overexpression of AAK-2/AMPK increases lifespan, which is dependent upon CRTC-1 repression. At the molecular level, AMPK phosphorylation of CTRC-1 results in its inactivation; thus, AMPK is a negative regulator of CRTC-1. Loss of AMPK signalling would therefore be predicted to de-repress CRTC-1. Interestingly, we found that the increased secretion of FLP-7mCherry in the *aak-2* mutants was suppressed by *crtc-1* removal (Fig. 6a,b), and was restored to levels just below that of the wild-type. In the *aak-2;crtc-1* double mutant, the suppression of FLP-7 secretion was accompanied by a full suppression of the reduced body fat seen in the *aak-2* single mutants (Fig. 6d). These data indicated that in wild-type animals, the presence of AAK-2/AMPK in the ASI neurons serves to keep CRTC-1 inactive, thus restraining FLP-7 secretion and limiting fat loss. We restored *crtc-1* cDNA in the ASI neurons in the *aak-2;crtc-1* double

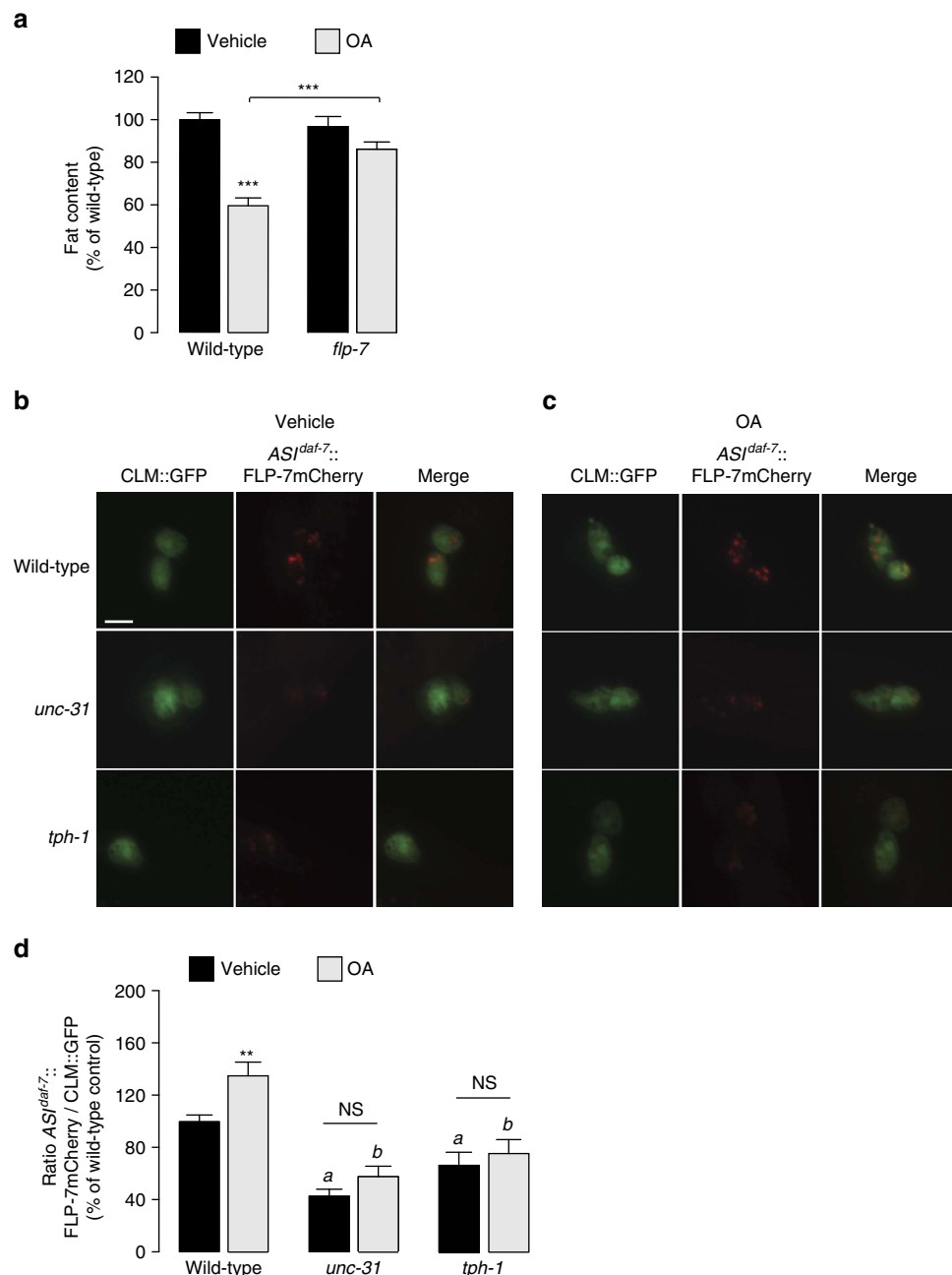

**Figure 5 | FLP-7 is required for OA-induced fat loss. (a)** Vehicle- or OA-treated wild-type and *flp-7* mutant animals were fixed and stained with oil Red O. Fat content was quantified for each genotype and is expressed as a percentage of vehicle-treated wild-type animals ± s.e.m. (lower panels; $n = 11$-24). ***$P < 0.001$ by two-way ANOVA. **(b,c)** Representative images for vehicle- and OA-treated wild-type, *unc-31*, and *tph-1* animals bearing integrated FLP-7mCherry and CLM::GFP transgenes. Left columns, GFP expression in coelomocytes; centre panels, secreted FLP-7mCherry uptake in coelomocytes; right panels (merge). Scale bar, 10 µm. **(d)** The intensity of FLP-7mCherry fluorescence was quantified and normalized to the area of the CLM::GFP. Data are expressed as a percentage of the normalized FLP-7mCherry fluorescence intensity of vehicle-treated wild-type animals ± s.e.m. (lower panels; $n = 8$-27). Within group analyses is represented by **$P < 0.01$; **(a)** $P < 0.05$ across the vehicle treatment groups (black bars); **(b)** $P < 0.001$ across the OA treatment groups (grey bars) and NS, not significant; two-way ANOVA used for the statistical comparisons.

mutants and observed a robust partial restoration of the increased FLP-7mCherry secretion (Fig. 6a,b), similar to that seen in the *aak-2* mutants alone. In wild-type animals, ASI-specific expression of a constitutively active form of *crtc-1* also increased FLP-7mCherry secretion (Fig. 6a,b). Together, our data suggest that in wild-type ASI neurons, AMPK signalling serves to keep the CREB co-regulator CRTC-1 inactive, which in turn restrains FLP-7 secretion. On the other hand, loss of AAK-2 or increased 5-HT signalling de-repress

CRTC-1, which stimulates FLP-7 release and fat loss. These data suggest a mechanism to couple nutrient information in the environment, with regulating fat metabolism. Food sensory information is relayed via the 5-HT circuit and the ASI neurons, which function as gatekeepers of the extent of body fat loss via regulating FLP-7 release. During food scarcity, AMPK in ASI neurons would become activated, thus repressing FLP-7 release and conserving body fat stores.

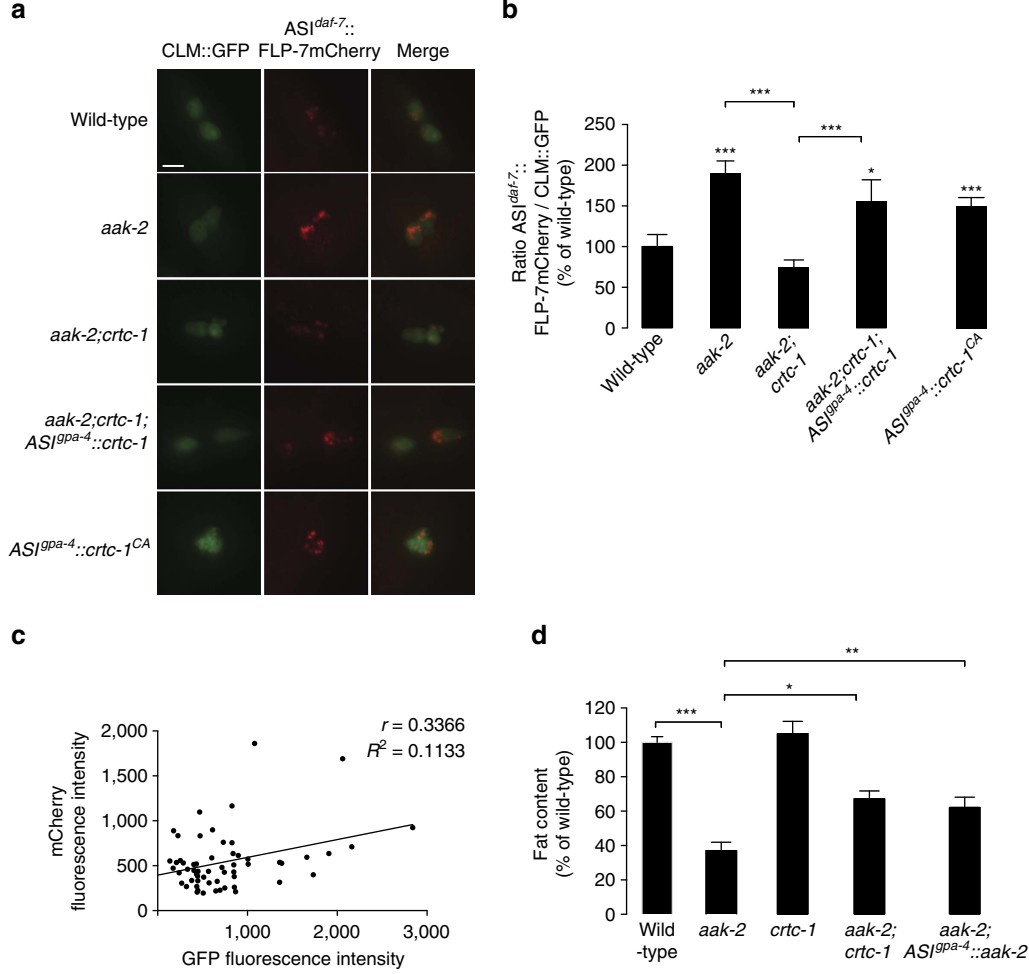

**Figure 6 | The nutrient sensor AAK-2/AMPK regulates FLP-7 release from ASI neurons. (a)** Representative images of wild-type, *aak-2*, *ctrc-1* and *aak-2;crtc-1* animals, with the indicated rescuing transgenes in the FLP-7mCherry and CLM::GFP secretion line. Left panels, GFP expression in coelomocytes; centre panels, secreted FLP-7mCherry uptake in coelomocytes; right panels (merge). Scale bar, 10 μm. **(b)** For vehicle- and 5-HT-treated animals bearing integrated FLP-7mCherry and CLM::GFP transgenes, the intensity of FLP-7mCherry fluorescence within a single coelomocyte was quantified and normalized to the area of CLM::GFP expression. Genotypes are indicated in the figure. Data are expressed as a percentage of the normalized FLP-7mCherry fluorescence intensity of vehicle-treated wild-type animals ± s.e.m. ($n = 10$–25 animals). *$P < 0.05$ and **$P < 0.01$, ***$P < 0.001$ by one-way ANOVA. **(c)** mCherry fluorescence intensity values are plotted against GFP fluorescence intensity values for each animal across representative experimental conditions, $n = 58$. **(d)** Fat content was quantified in wild-type animals *and aak-2, crtc-1, aak-2;crtc-1* mutants and *aak-2;crtc-1* mutants bearing a *aak-2* transgene the ASI-specific *gpa-4* promoter, as indicated. Data are expressed as a proportion of fat retained upon 5-HT treatment ± s.e.m. ($n = 18$–20). **$P < 0.01$ and ***$P < 0.001$ by two-way ANOVA.

**NPR-22/NK2R is the *in vivo* GPCR for FLP-7.** Next, we sought to identify the receptor through which FLP-7 acts. Two G protein coupled receptors called neuropeptide receptor-22 (NPR-22) and FMRFamide peptide-like receptor-3 (FRPR-3) had been previously shown to use FLP-7 peptides as ligands for Gq-mediated activation in HEK293 cells[44,45]. To obtain *in vivo* evidence for a FLP-7 GPCR, we tested *npr-22* and *frpr-3* null mutants for suppression of 5-HT-mediated body fat loss. *npr-22* mutants suppressed 5-HT-induced fat loss to the same extent as *flp-7* mutants, whereas *frpr-3* mutants did not, and resembled wild-type animals with respect to body fat with and without 5-HT (Fig. 7a). A polycistronic GFP reporter under the control of the *npr-22* promoter revealed expression in a few pairs of head neurons and in the intestine, but not in other tissues including muscle and hypodermis (Fig. 7b). We generated transgenic animals in the *npr-22* background by restoring *npr-22* expression under the control of its own promoter, pan-neuronally, and in the intestine. Relative to

non-transgenic *npr-22* mutants, transgenic animals bearing the endogenous *npr-22* promoter fully rescued 5-HT-mediated fat loss (Fig. 6c). Restoration of *npr-22* expression under a pan-neuronal promoter did not permit 5-HT-mediated fat loss in transgenic animals. On the other hand, re-expression of *npr-22* selectively in the intestine led to a full rescue of 5-HT-mediated fat loss (Fig. 6c). With respect to 5-HT-induced fat loss, transgenic animals bearing the rescuing transgene either under the endogenous *npr-22* promoter or under the intestine-specific promoter were indistinguishable from one another. In addition, *npr-22* and *flp-7;npr-22* mutants both fully suppressed the 5-HT-mediated induction of *atgl-1* transcription (Fig. 6d), suggesting that the neuronal 5-HTergic signal for fat loss is conveyed to the intestine via FLP-7/NPR-22 signalling.

NPR-22 is a *C. elegans* orthologue of the mammalian tachykinin/neurokinin 2 receptor (*TacR2* or NK2R; Supplementary Fig. 6). The mammalian NK2 receptor is expressed in many tissues but is enriched in the adrenal gland, the small

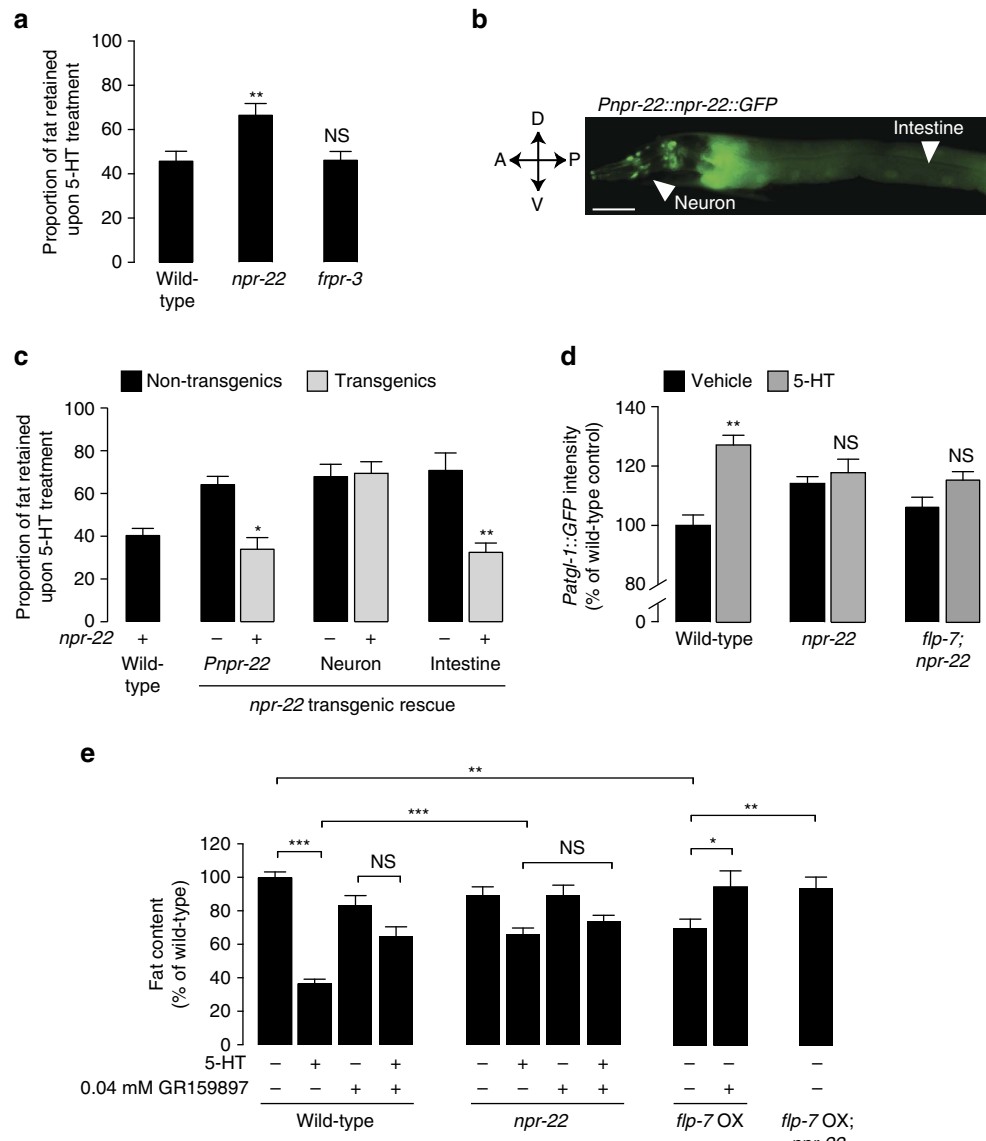

**Figure 7 | The GPCR NPR-22/Tachykinin 2 receptor (NK2R) functions as the FLP-7 receptor in the intestine. (a)** Fat content of vehicle- and 5-HT-treated wild-type, *npr-22* and *frpr-3* animals fixed and stained with oil Red O was quantified. Fat content for each genotype is expressed as a proportion of fat retained upon 5-HT treatment ± s.e.m. (lower panels; $n = 10$-13). **$P < 0.01$ and NS, not significant by two-way ANOVA. **(b)** Fluorescent image of a transgenic animal bearing an *npr-22::GFP* transgene under the control of the endogenous *npr-22* promoter. GFP expression was observed in the intestine and several pairs of neurons in the head. A, anterior; P, posterior; V, ventral; D, dorsal. Scale bar, 50 μm. **(c)** Fat content of vehicle- and 5-HT-treated wild-type animals and *npr-22* mutants fixed and stained with oil Red O was quantified, and is indicated as a proportion of fat retained upon 5-HT treatment. For the transgenic lines bearing *npr-22* expression, the promoters used are indicated, and non-transgenic animals are marked as ( − ) and transgenic animals as ( + ). The *unc-31* promoter was used for expression in neurons and the *ges-1* promoter was used for expression in the intestine. Data are expressed as a proportion of fat retained upon 5-HT treatment ± s.e.m. (lower panels; $n = 8$-12). *$P < 0.05$ and **$P < 0.01$ by two-way ANOVA. **(d)** The fluorescence intensity of *atgl-1* expression in vehicle- and 5-HT-treated wild-type animals and *npr-22* and *flp-7;npr-22* mutants bearing an integrated *atgl-1::GFP* transgene was quantified. The fluorescence intensity is expressed as a percentage of vehicle-treated wild-type animals ± s.e.m. ($n = 17$-25). **$P < 0.01$ and ns, not significant by two-way ANOVA. **(e)** Wild-type animals and *npr-22* mutants bearing the *flp-7* over-expression (OX) transgene were grown on plates containing either vehicle (10% dimethyl sulfoxide) or the selective NK2R antagonist GR159897 at the indicated concentration. At the completion of development (late L4 stage), animals were transferred to plates containing GR159897 and either vehicle or 5-HT. Fat content was quantified for each condition and is expressed as a percentage of vehicle-treated wild-type animals ± s.e.m. (11–20). *$P < 0.05$, **$P < 0.01$, ***$P < 0.001$ and NS, not significant by two-way ANOVA.

intestine, T cells and macrophages (www.biogps.org); however, no functional studies have previously associated it with lipid metabolism. To further evaluate the conservation between NPR-22 and the mammalian NK2 receptor, we took a pharmacological approach to block NK2 receptor activity using GR159897, a potent and selective antagonist of the

NK2 receptor[46]. Relative to mock treatment, pre-exposure to GR159897 effectively blocked 5-HT-induced fat loss in a dose-dependent manner in wild-type animals, with no observable developmental delays or lethality (Supplementary Fig. 7). To study the interaction between the mammalian NK2R antagonist and the *C. elegans* NPR-22 orthologue, we

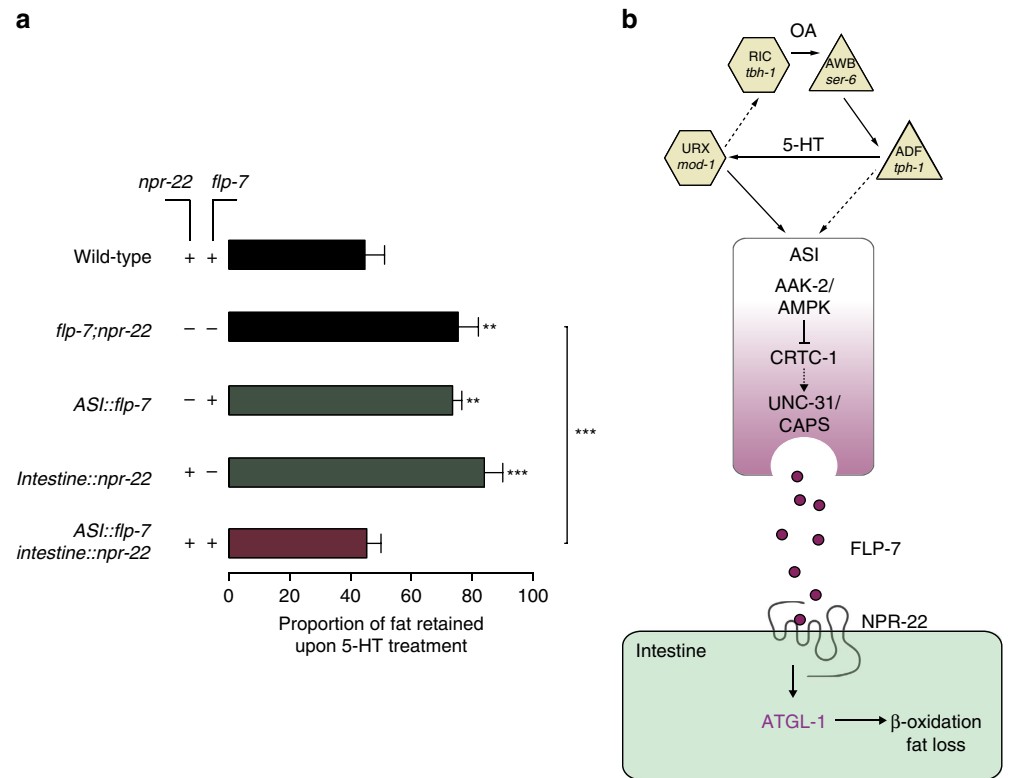

**Figure 8 | FLP-7 and NPR-22 function as a fat regulatory neuroendocrine ligand-receptor pair *in vivo*.** (**a**) Fat content was quantified in vehicle- and 5-HT-treated wild-type animals and in *flp-7;npr-22* double mutants bearing the *flp-7* and/or *npr-22* transgenes under the control of the ASI *daf-7* or intestinal *ges-1* promoters, as indicated. Data are expressed as a proportion of fat retained upon 5-HT treatment ± s.e.m. ($n = 11$–26). \*\**P* < 0.01 and \*\*\**P* < 0.001 by two-way ANOVA. (**b**) Model depicting the FLP-7/NPR-22 neuroendocrine axis that underlies the 5-HTergic control of body fat loss. In the nervous system, an integrated 5-HT and octopaminergic circuit stimulates body fat loss. In this study, we report the discovery of a tachykinin signalling system that underlies the 5-HTergic control of body fat loss in *C. elegans*. The FLP-7 neuroendocrine peptide is secreted from the ASI neurons in response to 5-HT and Oct-mediated signalling. The nutrient sensor AAK-2/AMPK acts in the ASI neurons via the CREB co-regulator CRTC-1 to regulate FLP-7 release in response to 5-HT-encoded signals of food availability. Upon release, FLP-7 acts in the intestine via the NPR-22/NK2R receptor to stimulate the ATGL-1 lipase, which drives fat loss. The identification of FLP-7 and NPR-22 addresses a long-standing question about the molecular basis of the central effects of 5-HT on fat loss in peripheral tissues.

first asked the following questions: to what extent does the antagonist compound mimic the suppressive effect of fat loss seen in the *npr-22* mutants, and is there an additive effect of NPR-22 and the compound together? As seen in Fig. 7e, in wild-type animals the fat loss elicited by 5-HT is suppressed nearly two-fold by pre-treatment with the GR159897 antagonist. *npr-22* mutants suppress 5-HT-induced fat loss to the same extent as GR159897, and treatment of *npr-22* mutants with the compound did not lead to an additive effect on the suppression (Fig. 7e), suggesting that the NK2R antagonist does not function via an independent receptor. Next, in the *flp-7* overexpression line (which drives fat loss via *atgl-1*; Fig. 1d), we found that both *npr-22* removal and NK2R antagonist treatment fully suppress the fat loss induced by increased FLP-7 signalling (Fig. 7e). These experiments show that the loss of the NPR-22 GPCR and the mammalian NK2R antagonist can interchangeably block the effects of 5-HT signalling and FLP-7 overexpression on fat loss in the intestine.

***In vivo* ligand-receptor necessity and sufficiency.** The relationship between the FLP-7 ligand and the NPR-22 receptor had been shown in cultured HEK293 cells[44]. To test ligand-receptor necessity and sufficiency *in vivo*, we asked the following question: in the context of a *flp-7;npr-22* double mutant, would either the ligand alone or the receptor alone suffice to restore 5-HTergic fat loss? We reasoned that if FLP-7 and NPR-22 are

a true ligand-receptor pair *in vivo*, neither gene could rescue the double mutant without the presence of the other. We crossed the neuronal *flp-7* transgenic rescue line with the intestinal *npr-22* rescue line. Because *C. elegans* transgenes are carried as extrachromosomal arrays, we simultaneously isolated *flp-7;npr-22* double mutants and those bearing either the *flp-7* transgene alone, the *npr-22* transgene alone, or both. First, we found that *flp-7;npr-22* double mutants suppressed 5-HT-induced fat loss to the same extent as either single mutant alone with no additive effects, suggesting that they function in a single linear pathway (Fig. 8a). Next, in the *flp-7;npr-22* mutant background, re-expression of either the *flp-7* transgene alone or the *npr-22* transgene in the intestine did not allow 5-HT-induced fat loss (Fig. 8a). In contrast, double mutants in which both the *flp-7* and *npr-22* transgenes were re-expressed in their respective locations (ASI neurons and intestine, respectively) showed a complete restoration of 5-HT-induced fat loss (Fig. 8a). Thus, we conclude that the FLP-7 ligand and the NPR-22 receptor function as a true ligand-receptor pair *in vivo*, and define the neuroendocrine axis that couples nutrient information via 5-HT signalling in the nervous system, with fat loss in the periphery.

**Discussion**

Here, we present the identification of a secreted neuroendocrine peptide called FLP-7 and its cognate *in vivo* GPCR NPR-22, the

ancestral orthologues of the tachykinin/neurokinin signalling system, as the neuroendocrine signalling axis that connects the 5-HT circuit from the nervous system, to its metabolic actions in the intestine. FLP-7 and NPR-22 function together as a ligand-receptor pair and define a conserved signalling pathway with an unexpected role in fat metabolism. We show that the FLP-7 peptide is secreted from neurons, and acts in the intestine via the tachykinin 2 receptor orthologue NPR-22 to mobilize fat stores (Fig. 8b). We find that FLP-7 is a potent and selective stimulator of body fat loss without altering 5-HT-dependent changes in food intake, locomotion or reproduction. Our studies presented here uncover a previously undefined neuroendocrine signalling pathway that preferentially stimulates body fat loss. We hypothesize that for global neuromodulators of physiology and behaviour such as 5-HT, one mechanism by which phenotypic selectivity is achieved is to deploy distinct neuropeptides for each phenotypic output. Thus, identification of selective regulators for distinct physiological outputs represents a path forward to uncover novel and potent neuroendocrine regulators of physiology.

Our identification of the ASI sensory neurons as the source of FLP-7 secretion is tantalizing. Several lines of evidence indicate a role for these neurons as gatekeepers of communication from the sensory nervous system to the metabolic organs of the body. In this regard, these neurons resemble the mammalian hypothalamus-brainstem complex. First, the ASI neurons are known to regulate both lifespan and dauer formation[33,34,47], an alternate physiological state that is triggered by unfavourable environmental conditions during development. Second, at least two distinct neuropeptides are already known to be secreted from ASI neurons: the TGFβ ligand DAF-7 (ref. 48), and the insulin DAF-28 (ref. 35). Although these neuropeptides do not have roles in 5-HTergic fat loss, they are nonetheless regulators of global physiological state. Third, ASI neurons regulate locomotor and feeding behaviours in response to food availability and pheromones[49–51]. Finally, ER stress-induced unfolded protein response and innate immunity pathways are activated in ASI neurons to promote dauer diapause[52,53]. The demonstration of a role for FLP-7 secretion from the ASI neurons in response to 5-HT signalling and the nutrient sensor AMPK suggests that these neurons integrate nutrient information to regulate multiple physiological outputs. We propose a model in which nutrient information from the environment regulates fat metabolism via this neuroendocrine pathway (Fig. 8b). Nutrient replete conditions that evoke increased neuronal 5-HT keep AAK-2/AMPK inactive in the ASI neurons, which de-represses CRTC-1, thus stimulating FLP-7 secretion and promoting fat loss via NPR-22 in the intestine (Fig. 8b). On the other hand, nutrient depletion activates AAK-2/AMPK signalling in the ASI neurons, represses CRTC-1, resulting in reduced FLP-7 secretion. Thus, fat loss is halted in the face of decreased food availability (Fig. 8b). One implication of our results is that limiting rapid fat loss and mitochondrial beta oxidation via modulation of FLP-7 secretion may be the principal mechanism underlying the increased lifespan of AMPK overexpression[43]. A major future challenge would be to decipher the relationship between nutrient sensing in the nervous system, and the mechanisms governing the secretion of each distinct neuropeptide from the ASI neurons to elicit context-dependent responses that sculpt the physiological response to metabolic and lifespan control.

Of the many neuropeptide genes encoded by the C. elegans genome, FLP-7 emerged as the predominant suppressor of 5-HT-mediated fat loss. The tachykinin family of peptides was discovered over 80 years ago with the extraction and isolation of its founding member Substance P, from horse intestine and

brain[54]. It is now known that the mammalian tachykinins are excitatory neuropeptides found predominantly in the hypothalamus and in the primary sensory neurons of the enteric nervous system and in the dorsal root, trigeminal and vagal ganglia that innervate the gastrointestinal tract[29,55]. The tachykinin 'brain-gut' peptides have many roles in mammalian physiology including gut motility, but have been studied primarily with respect to their roles in nociception and inflammation[56–58]. In mammals there are three tachykinin receptors (all GPCRs), called NK1R-NK3R (refs 59–61). NK1R is the best-studied receptor, is widely expressed, and NK1R antagonists are in clinical use as anti-emetics[62,63]. NK2R, the orthologue of the C. elegans NPR-22 identified in this study, is perhaps the most under-studied tachykinin receptor. It is expressed in several metabolic tissues in mice and humans including the intestine, pancreas, liver and kidney (www.biogps.org). To our knowledge, NK2R has not previously been associated with changes in adiposity or energy balance in mammals. It will be interesting to determine the extent to which central 5-HT action leads to stimulation and release of the tachykinin peptides from specific primary sensory neuron populations in other systems. In C. elegans, another neuropeptide called FLP-18 also regulates behaviour and metabolism through two GPCRs, NPR-4 and NPR-5. Unlike FLP-7, FLP-18 has broad roles in coordinating behaviour and metabolism across many tissues, suggesting a complex regulatory landscape for neuropeptide biology[64].

Other than the requirement for UNC-31/CAPS, the mechanisms of neuropeptide release have remained unknown, in any species. As in the case of the regulated release of synaptic vesicles[65], it is likely that there are many additional factors regulating the release of dense core vesicles. Identifying such regulators has the potential to reveal insights into many aspects of neuromodulation and neuroendocrine biology. Finally, despite recent strides in functionally characterizing the roles of neuropeptides in behaviour[24,66,67], the great majority of neuropeptides in metazoan genomes remain unannotated. We suggest that the many non-cell-autonomous physiological effects described in the C. elegans literature stem from neuropeptides that function as endocrine factors[68–71]. The approach presented here could be one useful strategy to uncover novel facets of neuroendocrine biology.

## Methods

**Animal maintenance and strains.** Nematodes were cultured on OP50 bacterial lawns on nematode growth medium (NGM) plates at 20 °C (ref. 72). N2 Bristol, obtained from the Caenorhabditis Genetic Center (CGC), was used as the wild-type reference strain. The strains used in the study are given in Supplementary Table 1. The list of neuropeptide mutants screened is available on request. With the exception of behavioural assays, all animals were synchronized by hypochlorite treatment, after which hatched L1 larvae were seeded on OP50 plates. All experiments were performed on day 1 adults.

**5-HT OA and GR159897 treatments.** 5-HT hydrochloride powder (Sigma) and octopamine hydrochloride (OA; Sigma) were dissolved in 0.1 M HCl. Both 5-HT and OA were freshly added to plates, and used at a final concentration of 5 mM (ref. 1). GR159897 (Tocris) was used at the final concentrations of 0.04, 0.016 and 0.008 mM in 10% dimethyl sulfoxide. The doses were chosen after ensuring that there were no developmental delays, or other overt effects on growth.

**Oil Red O staining.** Nematodes were harvested with phosphate-buffered saline (PBS) and incubated on ice for 10 min. Animals were fixed in 60% isopropanol on a rotating rack for 20 min. Following this step, animals were incubated overnight with a filtered oil Red O working solution (60% oil Red O in isopropanol: 40% water) and protected from light. Within a single experiment, approximately 2,000 animals were fixed and stained. Within each experiment, ~100 animals were visually inspected on slides, following which 15–20 animals

were imaged for each genotype/condition. All experiments were repeated at least three times.

**Image acquisition and quantitation.** Black and white images of oil Red O stained animals and fluorescent images were captured using a ×10 objective on a Zeiss Axio Imager microscope. In all cases, lipid droplet staining in the first four pairs of intestinal cells was quantified using NIH Image J software, by measuring pixel intensity over the first four pairs of intestinal cells, which fully encapsulate the variation seen in the tested conditions[1]. For all *atgl-1::GFP* images, fluorescence intensity in the second and third pairs of intestinal cells was quantified. Fluorescent images of the reporters for FLP-7 secretion were captured using a ×20 objective on a Zeiss Axio Imager microscope. For all 'secretion line' animals, the first pair of coelomocytes was imaged. mCherry fluorescence intensity in one of the two imaged coelomocytes was quantified and normalized to the surface area of the coelomocyte. Within each experiment, approximately 15–20 animals from each condition were quantified. All images were quantified using ImageJ (NIH).

**Cloning and transgenic construction.** Promoters and genes were cloned using standard PCR techniques from N2 Bristol lysates, using Gateway Technology (Life Technologies). Promoter lengths were determined based on functional rescue and are available on request. All rescue plasmids were generated using polycistronic GFP. Transgenic rescue strains were constructed by microinjection into the *C. elegans* germline followed by visual selection of transgenic animals under fluorescence. For the microinjections, 5–10 ng μl$^{-1}$ of the desired plasmid was injected with 25 ng μl$^{-1}$ of an *unc-122::GFP* co-injection marker and 65–70 ng μl$^{-1}$ of an empty vector to maintain a total injection mix concentration of 100 ng μl$^{-1}$. In each case, 10–20 stable transgenic lines were generated. Two lines were selected for experimentation based on consistency of expression and transmission rate. The FLP-7 secretion lines were generated with plasmid containing a *flp-7::mCherry* fusion construct under the control of the *daf-7* and *str-3* promoters.

**Quantitative RT-PCR.** Total RNA was extracted using TRIzol reagent (Invitrogen). Genomic DNA was removed using an RNase-free DNase kit (QIAGEN). cDNA was prepared using a iScript Reverse Transcription Supermix for RT-qPCR kit (BioRad) according to the manufacturer's instructions. Quantitative PCR was performed using the SsoAdvanced Universal SYBR Green Supermix according to the manufacturer's instructions. Data were normalized to actin mRNA. Primer sequences are listed in Supplementary Table 2.

**DiI staining.** Animals of mixed developmental stages were incubated in a 1:200 dilution of DiI stain (Life Technologies) overnight on a rotating rack. After staining, the animals were seeded onto a plate containing an OP50 bacterial lawn and allowed to dry for approximately 30 min. Fluorescent images of animals in the L2-L3 larval stages were captured using a ×100 objective on a Zeiss Axio Imager microscope.

**Food intake.** Food intake was measured using feeding rate[11]. For each animal, the rhythmic contractions of the pharyngeal bulb were counted over a 10 s period under a Zeiss M2 Bio Discovery microscope. For each genotype, ten animals were counted per condition and the experiment was repeated at least three times.

**Egg-laying assay.** Animals were treated with 5 mM 5-HT or vehicle for 46 h at 20 °C. For each condition, five animals were then transferred onto 3.5 cm plates containing 5-HT or vehicle, respectively. For each condition, a total of ten plates were assayed. After 3 h the animals were removed from the experimental plates and the number of eggs on the plate was counted. Data are presented as the average number of eggs laid per hour.

**Enhanced slowing response.** The enhanced slowing response was assayed as described[6]. Day 1 adult animals were washed off food plates with PBS, washed five times to remove food and placed on NGM agar plates without food. After a 30-min fast, animals were collected in PBS and seeded onto NGM plates with HB101. Animals were allowed to acclimatize for 5 min, after which the number of body bends/20 s was counted.

**Statistics.** Wild-type animals were included as controls for every experiment. Error bars represent s.e.m. Student's *t*-test, one-way ANOVA and two-way ANOVA were used as indicated in the figure legends.

**Data availability.** The authors confirm that all relevant data are included in the paper and/or its supplementary information files, or can be obtained from the author on reasonable request.

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

## Acknowledgements

This work was supported by research grants to S.S. from the NIH/NIDDK (R01 DK095804). We are grateful to the Knockout Consortium at Tokyo Women's Medical University for strains. Some strains were provided by the CGC, which is funded by NIH Office of Research Infrastructure Programs (P40 OD010440). We thank Dr William Mair (Department of Genetics and Complex Diseases, Harvard University) for the *crtc-1* mutant strain and crtc-1 cDNA. We thank Dr Rosalind Hussey and members of the Srinivasan Lab for critical comments on the manuscript.

## Author contributions

S.S., L.P. and T.N. designed the study. L.P. and T.N. conducted the experiments with contributions from E.W., H.R. and M.V. S.S. and L.P. analysed the data. S.S. and L.P. wrote the manuscript. All authors read and approved the manuscript.

## Additional information

**Competing financial interests:** The authors declare no competing financial interests.

