## [Peer Review File · Nature Communications]

Reviewer #1 (Remarks to the Author)

In the manuscript "A tachykinin-mediated neuroendocrine axis couples central serotonin action with peripheral lipid metabolism", Srinivasan and collaborators identify a secreted neuropeptide (FLP-7) and its receptor (NPR-22) as a circuit that connects the 5-HT neuronal network to its impact on lipid metabolism in the gut.

Overall, this study is rigorous, sound and convincing and there is a significant amount of work that leads to interesting mechanistic insights. However, the novelty of this study is not that great because the tachykinin/lipid metabolism has already been described in fly by the Perrimon lab and the *npr-22/flp-7* interaction was already known. Providing additional experiments in mammals would definitely make this study a much better candidate for Nature Communications.

Comments:

1. The result (or a summary) of the screen performed to identify *flp-7* should be shown.
2. Are the *flp-7* transgenic lines used for secretion studies rescuing the null? It is somewhat unclear to me. Is the tag affecting functionality?
3. *flp-7* mRNA levels should also be measured in *tph-1* and *mod-5* mutants to ensure that secretion is affected, not transcription.
4. What are the mRNA levels of *flp-7* in the OE lines that exhibit decreased fat content? Did the authors generate several OE lines? Does the decrease correlate with *flp-7* transcript levels?
5. In the *flp-7* OE lines, is the fat decrease linked to ATGL-1? In other words, if one inhibits *atgl-1* by RNAi (this should decrease fat content), does *flp-7* OE still reduce fat? And if so to what extent?

Minor comment:

1. On page 13, there is an unfinished sentence that should be removed: "GR-159897 pretreatment"

Reviewer #2 (Remarks to the Author)

This study identifies a previously undescribed neuroendocrine signaling pathway that mediates the effects of serotonin on intestinal fat content in *C. elegans*. The authors show that mutants lacking the neuropeptide-like protein *flp-7* have defects in serotonin-induced reductions in intestinal Oil Red O staining and increases in the expression of an intestinal lipase gene. They identify the neuron ASI as a cellular source of *flp-7* and they show that *flp-7* acts through an intestinal GPCR, *npr-22*, to regulate fat content. They go on to examine the effects of chronic exposure to serotonin and of mutants defective in endogenous serotonin signaling on the efficacy of FLP-7 secretion from neurons. This study suggests that neuroendocrine signaling may be an evolutionarily conserved mechanism by which serotonin regulates metabolism in distal tissues. The identification of the specific peptide-GPCR pair and their sites of action is likely to provide an excellent system to study mechanism underlying neuronal control of fat regulation in the context of a multicellular organism. This study is not entirely conceptually novel since prior work by the authors suggested the existence of an endocrine mechanism of fat regulation, and other groups have previously identified other neuropeptide/GPCR pathways that regulate fat content through neuron-to-intestine signaling (e.g. deBono, 2009). A major concern with the paper lies in the interpretation of a central thesis of the paper: that FLP-7 secretion is serotonin-regulated (Figures 3 and 4). More experiments would need to be done to clarify whether the authors are witnessing serotonin-mediated regulation of FLP-7 secretion (as the authors conclude) or serotonin-mediated regulation of transgene expression.

Concerns:

- 1) Figs 3-4. The authors intended to examine *flp-7* secretion by generating transgenic animals expressing *flp-7mCherry* under control of the *daf-7* promoter. Many prior studies have shown that *daf-7* expression is regulated by food, starvation and serotonin itself. Specifically, *tph-1* mutations dramatically reduce expression from a *daf-7* promoter in ASI (Ruvkun, Nature, 2000). The authors showed that there is a significant reduction in coelomocyte fluorescence in *tph-1* mutants, but it is

difficult to tease out whether this reduction is due to a reduction in flp-7 expression from the daf-7 promoter or a decrease in the regulated release of FLP-7. Indeed all of the experiments in which serotonin levels were altered are difficult to interpret due to potential transcriptional effects of serotonin on the daf-7 promoter. The authors present the following control: "First, we did not observe any changes in the expression levels of flp-7 mRNA following 5-HT treatment (Supplementary Fig. 2a), therefore the observed increase in FLP-7mCherry punctae cannot be attributed to transcriptional effects". But here I believe they examined endogenous flp-7 expression, so this control is not relevant. The authors would need to perform the coelomocyte experiments using a different ASI promoter whose expression is not regulated by serotonin in order to distinguish effects of serotonin mutants on flp-7 expression vs FLP-7 secretion. Without this, it is impossible to determine whether the release of flp-7 is regulated by serotonin.

2) Abstract: "We report that in *C. elegans* the ancestral ortholog of the tachykinin peptides called FLP-7, and the tachykinin-2 receptor ortholog NPR-22...", and page 13 "NPR-22 is the *C. elegans* ortholog of the mammalian tachykinin/neurokinin 2 receptor (TacR2 or NK2R; Supplementary Fig. 5)." The orthology between FLP-7/NPR-22 and tachykinin signaling is overstated here and throughout the paper. NPR-22 is among multiple *C. elegans* GPCRs that share similar, if not more homology with NK2R than NPR-22. Likewise, there are other peptides that are more similar to tachykinin (eg. NLP-8) than FLP-7. The sequence alignment of FLP-7 shown in Supplemental figure 1 shows identity with the KR dipeptide which is presumably cleaved. Without the KR, similarity would not be convincing. Thus, referring to FLP-7 as tachykinin or NPR-22 to NK2R is not justified and may be misleading. The lack of strict orthology to tachykinin signaling components is not viewed as a weakness by this reviewer as the upstream and downstream signaling mechanisms in this pathway are likely to be conserved in higher organisms even if the signal itself is not.

2) All fat figures: The paper relies exclusively on the Oil Red O staining for analysis of intestinal fat content. There is some disagreement in the field as to what extent the Oil Red O assay is quantitative. Given the rather small changes in staining reported (~20% for flp-7), would it be possible to utilize an independent assay, like GCMS or Nile red for some of the more salient mutants to confirm that the changes in staining reported reflect changes in fat content?

3) Fig 2. The site of action of FLP-7 remains a bit unclear. In supplemental fig1c Flp-7 appears to be expressed in ASI, ALA, AVG as well as several other neurons. A daf-7 promoter fragment was used to knock down flp-7. The authors state that daf-7 is an ASI promoter, but daf-7 is not exclusively expressed in ASI. This raises the concern that the rescue and knock-down results shown may be due to expression in these other daf-7-expressing cells. This would be important to sort out experimentally because the proper identify of the cell is critical for the authors model.

5) Page 13. The issue of whether GR159897 is an npr-22 inhibitor is not clear. First, GR159897 treatment appears to reduce fat content in the absence of 5HT, a phenotype not shared with flp-7. The authors do not report the fat content of untreated npr-22 animals so it is impossible to compare the effects of GR159897 with npr-22 mutants. Second, "There was no additive suppressive effect of the compound in npr-22 mutants, indicating that antagonism of NPR-22/NK2R blocks 5-HT-induced fat loss..." The lack of additivity is not surprising because both conditions nearly completely block the ability of 5HT to reduce fat-i.e. this is a phenotype where additivity is impossible to observe. Perhaps a more meaningful test of the target of GR159897 would be to determine if GR159897 and npr-22 mutations can block the reduction in Oil Red O associated with flp-7 overexpression.

Minor comments:

1) Fig 4 showed that mod-1 mutants also have defect in FLP-7 secretion, does this indicate that URX neurons also function upstream of and regulates ASI neurons activity in lipid metabolism, or could it possible that URX neurons release another signal that regulates FLP-7 secretion in ASI neurons? The model puts URX in parallel but not upstream of ASI.

- 2) Does 5HT treatment rescue the defect in FLP-7 secretion of tph-1 mutants?
- 3) Fig 6.e showed npr-22 blocked atgl-1 expression, what effect does flp-7;npr-22 double mutant have on atgl-1 expression?
- 4) Fig 3.d needs a legend to show what black and grey bars mean.
- 5) Page 13: sentence fragment: "GR-159897 pre-treatment did not lead to at the doses tested."
- 6) Could the authors comment on how FLP-7 signaling may function with flp-18 (deBono, 2009) signaling to regulate fat content?

Reviewer #3 (Remarks to the Author)

Here, Palamiuc, et al. identify a ligand-receptor pair that mediates the serotonergic control of fat metabolism, and identify the sites of ligand expression (ASI) and receptor activity (intestine). Overall, I think that this is a strong paper, so I will limit my comments to the most important points:

Major:

1. What regulates flp-7 expression in ASI?
2. What role, if any, does CRTC-1 play in this pathway?

Addressing the transcriptional regulation of these factors is important to put it into the context of other known regulatory pathways.

Minor:

1. Figure 1A seems out of place, since it summarizes published work rather than the work done here.
2. Was the 2008 screen mentioned done using Oil Red O or Nile Red? If the latter, do the results stand?
3. were all of the flp/nlp/ins neuropeptides tested, or only a fraction? Important to state, even if flp-7 turned out to account for all of the activity (in case of redundancy).

Point-by-point response to reviewers' comments

Overall, the reviewers and editors found our work rigorous, sound, convincing and indicated that it was an excellent system in which to study the mechanisms underlying the neuronal control of fat metabolism. They also suggested that the work has led to interesting mechanistic insights. We have addressed the experimental concerns from all three reviewers as well as the editorial comments, added further information for clarity and made the relevant changes to the text and figures. Importantly, we have also extended the work to define a novel connection between nutrient-sensing and lifespan-extending pathways in the nervous system, and their role in regulating the release of neuroendocrine signals that control body fat stores in the periphery. As a result, the manuscript has strengthened.

Reviewer 1

In the manuscript "A tachykinin-mediated neuroendocrine axis couples central serotonin action with peripheral lipid metabolism", Srinivasan and collaborators identify a secreted neuropeptide (FLP-7) and its receptor (NPR-22) as a circuit that connects the 5-HT neuronal network to its impact on lipid metabolism in the gut.

Overall, this study is rigorous, sound and convincing and there is a significant amount of work that leads to interesting mechanistic insights. However, the novelty of this study is not that great because the tachykinin/lipid metabolism has already been described in fly by the Perrimon lab and the npr-22/flp-7 interaction was already known. Providing additional experiments in mammals would definitely make this study a much better candidate for Nature Communications.

Comments:

1. The result (or a summary) of the screen performed to identify flp-7 should be shown.

We screened all members of the neuropeptide and insulin gene families for which mutants exist (77/113). We have provided a written summary of the screen at the beginning of the results section and provided representative images of the screen data for each neuropeptide class in Figure 1b.

2. Are the flp-7 transgenic lines used for secretion studies rescuing the null? It is somewhat unclear to me. Is the tag affecting functionality?

Different lines were used for the *flp-7* transgenic rescue and secretion studies. In the rescue experiments (Figure 2), we restore *flp-7* expression in the *flp-7* null background, using a polycistronic element (*SL2* for *C. elegans*) that simultaneously drives expression of *flp-7* and *mCherry* as separate mRNAs in the relevant neurons. In these lines, we can visualize where *flp-7* is synthesized by using the mCherry reporter but we cannot visualize FLP-7 protein secretion since there is no tag on the *flp-7* cDNA. Therefore, there is also no interference of the tag with respect to functionality.

On the other hand, in the secretion experiments, we express FLP-7 as a fusion protein with mCherry from the ASI neurons. This line reports on secretion of the FLP-7mCherry fusion protein, and is crossed into various mutants (Figures 3, 4, 5 and 6) to assess neuronal mechanisms regulating secretion, but we do not measure the effect of the fusion protein on downstream metabolic processes because FLP-7 is a ligand whose functionality would likely be altered by the presence of the mCherry tag.

In response to a comment by Reviewer 2 (point 1), we also confirmed FLP-7 secretion and function from the ASI neurons using an independent promoter and a second set of transgenic rescues and secretion assays (Figure 2 and Supplementary Figure 3). These experiments have confirmed and reinforced our results.

3. *flp-7* mRNA levels should also be measured in *tph-1* and *mod-5* mutants to ensure that secretion is affected, not transcription.

We have added these controls in Supplementary Figure 3, and there is no effect of transcription; in these genotypes *flp-7* mRNA levels are not different from wild-type.

4. What are the mRNA levels of *flp-7* in the OE lines that exhibit decreased fat content? Did the authors generate several OE lines? Does the decrease correlate with *flp-7* transcript levels?

We generated two overexpression lines for *flp-7*. In both cases, relative to the non-transgenic controls, we observed decreased body fat content. In *C. elegans*, transgenes are carried as extrachromosomal arrays that are not integrated within the genome. Therefore, a single transgenic parent will generate both non-transgenic and transgenic offspring. Isolating sufficient transgenic animals for qPCR would require hand-picking several thousand worms within a short period (generation time is 2-3 days). To circumvent this significant hurdle, we instead measure the fluorescence intensity of a reporter that is expressed from a polycistronic element, driven by the same promoter as *flp-7*. In both overexpression lines we generated, observed similar fluorescence intensities and a concomitant similar (and not significantly different from one another) decrease in body fat stores. The data from both lines is shown below, however for simplicity we have only included the data from line #1 in the manuscript (Figure 1d).

5. In the *flp-7* OE lines, is the fat decrease linked to *ATGL-1*? In other words, if one inhibits *atgl-1* by RNAi (this should decrease fat content), does *flp-7* OE still reduce fat? And if so to what extent?

We have conducted the suggested experiment and included the data in Figure 1d. Inactivation of *atgl-1* by RNAi suppresses fat loss to approximately the same extent in the overexpression line as it does in wild-type animals.

Minor comment:

1. On page 13, there is an unfinished sentence that should be removed: "GR-159897 pretreatment"

Removed.

Reviewer 2

This study identifies a previously undescribed neuroendocrine signaling pathway that mediates the effects of serotonin on intestinal fat content in C. elegans. The authors show that mutants lacking the neuropeptide-like protein flp-7 have defects in serotonin-induced reductions in intestinal Oil Red O staining and increases in the expression of an intestinal lipase gene. They identify the neuron ASI as a cellular source of flp-7 and they show that flp-7 acts through an intestinal GPCR, npr-22, to regulate fat content. They go on to examine the effects of chronic exposure to serotonin and of mutants defective in endogenous serotonin signaling on the efficacy of FLP-7 secretion from neurons. This study suggests that neuroendocrine signaling may be an evolutionarily conserved mechanism by which serotonin regulates metabolism in distal tissues. The identification of the specific peptide-GPCR pair and their sites of action is likely to provide an excellent system to study mechanism underlying neuronal control of fat regulation in the context of a multicellular organism. This study is not entirely conceptually novel since prior work by the authors suggested the existence of an endocrine mechanism of fat regulation, and other groups have previously identified other neuropeptide/GPCR pathways that regulate fat content through neuron-to-intestine signaling (e.g. deBono, 2009). A major concern with the paper lies in the interpretation of a central thesis of the paper: that FLP-7 secretion is serotonin-regulated (Figures 3 and 4). More experiments would need to be done to clarify whether the authors are witnessing serotonin-mediated regulation of FLP-7 secretion (as the authors conclude) or serotonin-mediated regulation of transgene expression.

Comments:

1. *Figs 3-4. The authors intended to examine flp-7 secretion by generating transgenic animals expressing flp-7mCherry under control of the daf-7 promoter. Many prior studies have shown that daf-7 expression is regulated by food, starvation and serotonin itself. Specifically, tph-1 mutations dramatically reduce expression from a daf-7 promoter in ASI (Ruvkun, Nature, 2000). The authors showed that there is a significant reduction in coelomocyte fluorescence in tph-1 mutants, but it is difficult to tease out whether this reduction is due to a reduction in flp-7 expression from the daf-7 promoter or a decrease in the regulated release of FLP-7. Indeed all of the experiments in which serotonin levels were altered are difficult to interpret due to potential transcriptional effects of serotonin on the daf-7 promoter. The authors present the following control: "First, we did not observe any changes in the expression levels of flp-7 mRNA following 5-HT treatment (Supplementary Fig. 2a), therefore the observed increase in FLP-7mCherry punctae cannot be attributed to transcriptional effects". But here I believe they examined endogenous flp-7 expression, so this control is not relevant. The authors would need to perform the coelomocyte experiments using a different ASI promoter whose expression is not regulated by serotonin in order to distinguish effects of serotonin mutants on flp-7 expression vs FLP-7 secretion. Without this, it is impossible to determine whether the release of flp-7 is regulated by serotonin.*

We conducted the following experiments to evaluate the suggestion that the observed FLP-7mCherry secretion effects were a function of the *daf-7* promoter. **(i)** In the *ASI^{daf-7}::flp-7mCherry* secretion line, we measured *flp-7mCherry* mRNA in wild-type, *tph-1*, *mod-5* and 5-HT-treated animals. We did not observe any significant differences in mRNA levels across these genotypes and conditions (Supplementary Figure 3a). **(ii)** We measured endogenous *daf-7* expression in wild-type, *tph-1*, *mod-5* and 5-HT-treated animals. As the reviewer points out, we did observe a decrease in *daf-7* mRNA expression in *tph-1* mutants, but not in the *mod-5* or 5-HT-treated conditions (Supplementary Figure 3b). **(iii)** Additionally, we examined the canonical *daf-7::GFP* reporter line and did not observe a difference in GFP fluorescence in the ASI neurons upon 5-HT treatment (Figure S3c).

Taken together, our data show that although endogenous *daf-7* expression is regulated by *tph-1* (but not by 5-HT or *mod-5*), *flp-7mCherry* mRNA levels remain independent of the *daf-*

7 promoter, possibly due to overexpression and integration of the *flp-7mCherry* transgene, or due to promoter-independent effects of *daf-7* mRNA regulation, potentially from the 3'UTR. Therefore, we believe that our results stand as described, and that FLP-7mCherry coelomocyte uptake using our transgenic line can be used to assess FLP-7 secretion.

However, to fully alleviate this concern, we used a second ASI-specific promoter to verify our results. We used the *str-3* promoter, which is not known to be regulated by 5-HT or *tph-1*^{1,2}. Using this promoter, we generated: (i) new rescue lines to evaluate localization of *flp-7* synthesis, and (ii) new secretion lines to evaluate FLP-7 secretion from the ASI neurons in different genotypes and conditions. We found that *ASI^{str-3}::flp-7* transgenics restore 5-HT-induced fat loss to the same extent as the *flp-7* and *daf-7* promoters (Figure 2b). Thus, we confirm the site of *flp-7* synthesis as the ASI neurons. We also examined FLP-7mCherry secretion using the *ASI^{str-3}* promoter. The newly-generated secretion line (in the wild-type background) was crossed into the *mod-5* mutants. We did not cross this line into *tph-1* mutants because the baseline level of FLP-7mCherry secretion driven by the *str-3* promoter in wild-type animals is far lower than in the case of the *daf-7* promoter (Supplementary Figure 3d), and the results with the *tph-1* mutants would therefore be un-interpretable. As seen with the *ASI^{daf-7}* line, we found that increase in exogenous or endogenous 5-HT via loss of *mod-5* resulted in increased FLP-7 secretion from the ASI neurons. Additionally, between the two lines, the extent of the modulation of FLP-7 secretion by 5-HT and *mod-5* was similar (Figure 3b-d and Supplementary Figure 3d,e). The data regarding this point are presented in Figures 2 and Supplementary Figures 3 and 5. We conclude that FLP-7 secretion from ASI neurons is regulated in proportion to changes in neuronal 5-HT signaling.

2. Abstract: *"We report that in C. elegans the ancestral ortholog of the tachykinin peptides called FLP-7, and the tachykinin-2 receptor ortholog NPR-22...", and page 13 "NPR-22 is the C. elegans ortholog of the mammalian tachykinin/neurokinin 2 receptor (TacR2 or NK2R; Supplementary Fig. 5)." The orthology between FLP-7/NPR-22 and tachykinin signaling is overstated here and throughout the paper. NPR-22 is among multiple C. elegans GPCRs that share similar, if not more homology with NK2R than NPR-22. Likewise, there are other peptides that are more similar to tachykinin (eg. NLP-8) than FLP-7. The sequence alignment of FLP-7 shown in Supplemental figure 1 shows identity with the KR dipeptide which is presumably cleaved. Without the KR, similarity would not be convincing. Thus, referring to FLP-7 as tachykinin or NPR-22 to NK2R is not justified and may be misleading. The lack of strict orthology to tachykinin signaling components is not viewed as a weakness by this reviewer as the upstream and downstream signaling mechanisms in this pathway are likely to be conserved in higher organisms even if the signal itself is not.*

We agree with the reviewer that the orthology with FLP-7 and tachykinin is still an open question. However, reciprocal blast searches between NPR-22 and human NK2R yield a score of 1e-37 with 33% identity. We have clarified the issue of antagonism with the mammalian NK2R antagonist GR159897 (point 5; Figure 7 in the text) on NPR-22, and believe that these experiments have strengthened the case for NPR-22 as a functional NK2R ortholog. We have tempered the statements about FLP-7/tachykinin throughout the text.

3. All fat figures: *The paper relies exclusively on the Oil Red O staining for analysis of intestinal fat content. There is some disagreement in the field as to what extent the Oil Red O assay is quantitative. Given the rather small changes in staining reported (~20% for flp-7), would it be possible to utilize an independent assay, like GCMS or Nile red for some of the more salient mutants to confirm that the changes in staining reported reflect changes in fat content?*

In our published work on lipid regulation in *C. elegans*^{3, 4, 5} we have provided extensive verification of our fat staining methods, and have cross-referenced each method with the others. In addition to our own work, peer-reviewed, published reports from several *C. elegans* laboratories have rigorously validated the use of Oil Red O staining to measure body fat^{6, 7, 8, 9, 10, 11, 12}. The clear consensus is that Oil Red O staining, performed correctly, represents the true triglyceride lipid droplet compartment in the worm. Because fixation-based staining methods allow visualization of fat stores in different tissues, they provide additional spatial information not obtainable by triglyceride extraction, LC-MS or GC-MS-based methods. In addition, biochemical extraction procedures from large numbers of worms can mask the true intestinal lipid content because of lipid stores present in the eggs, which are not subject to metabolic regulation. In our previous papers on 5-HT signaling^{3, 4} and in unpublished work, we have extensively verified the critical 5-HT pathway mutants using: **(i)** lipid extraction followed by thin-layer chromatography and GC-MS; **(ii)** lipid extraction followed by LC-MS and targeted triglyceride composition and quantification; **(iii)** lipid extraction followed by triglyceride hydrolysis assays using commercial kits (Bioassay Systems™); **(iv)** Sudan Black B staining; **(v)** Nile Red staining and **(vi)** fixation-based Oil Red O staining. For the 5-HT pathway (and other published and unpublished mutants from our lab), fixation-based Oil Red O staining correlates very well with triglyceride levels, and has a dynamic range that reflects triglyceride levels.

Thus, the paper utilizes Oil Red O based on a body of extensive and rigorous experimentation that show that fixation-based Oil Red O staining is the most quantitative and reliable method, in our laboratory, to measure intestinal fat stores. Our detailed method is published in Noble et al. 2013, and the Methods section now contains more information on the experimental methodology we follow. We have also included several more representative images of the *flp-7* phenotype in Supplementary Figure 1a.

4. Fig 2. The site of action of FLP-7 remains a bit unclear. In supplemental fig1c Flp-7 appears to be expressed in ASI, ALA, AVG as well as several other neurons. A daf-7 promoter fragment was used to knock down flp-7. The authors state that daf-7 is an ASI promoter, but daf-7 is not exclusively expressed in ASI. This raises the concern that the rescue and knock-down results shown may be due to expression in these other daf-7-expressing cells. This would be important to sort out experimentally because the proper identify of the cell is critical for the authors model.

Please see point 1 above. We have conducted the transgenic rescue experiments using another ASI promoter, *str-3*. The results are the same as with the *daf-7* promoter.

5. Page 13. The issue of whether GR159897 is an npr-22 inhibitor is not clear. First, GR159897 treatment appears to reduce fat content in the absence of 5HT, a phenotype not shared with flp-7. The authors do not report the fat content of untreated npr-22 animals so it is impossible to compare the effects of GR159897 with npr-22 mutants. Second, "There was no additive suppressive effect of the compound in npr-22 mutants, indicating that antagonism of NPR-22/NK2R blocks 5-HT-induced fat loss..." The lack of additivity is not surprising because both conditions nearly completely block the ability of 5HT to reduce fat-i.e. this is a phenotype where additivity is impossible to observe. Perhaps a more meaningful test of the target of GR159897 would be to determine if GR159897 and npr-22 mutations can block the reduction in Oil Red O associated with flp-7 overexpression.

We have taken the reviewer's suggestion and conducted the experiment in which we measure the effect of GR159897 and *npr-22*, on *flp-7* overexpression. As shown in Figure 7e, the *npr-22* mutant and the mammalian NK2R antagonist suppress 5-HT-mediated fat loss to the same extent, with no additive effect of the two combined. In either case the suppression is ~80% of wild-type, therefore had there been an additive effect, it would have been possible to visualize and quantify. On the other hand, as expected, the effect of *flp-7* overexpression on fat

loss was fully suppressed by *npr-22* and GR159897 (Figure 7e). Therefore, we believe that GR159897 likely functions as an NPR-22 inhibitor *in vivo*. We have now also included the *npr-22* fat phenotype on its own.

Minor comments:

1. Fig 4 showed that *mod-1* mutants also have defect in FLP-7 secretion, does this indicate that URX neurons also function upstream of and regulates ASI neurons activity in lipid metabolism, or could it possible that URX neurons release another signal that regulates FLP-7 secretion in ASI neurons? The model puts URX in parallel but not upstream of ASI.

Our data suggest that URX neurons are functioning upstream of the ASI neurons. However, at present we cannot formally exclude an additional role for a URX-independent route of communication from ADF to ASI, although we have not observed *mod-1* or *ser-6* expression in ASI neurons in the respective transgenic rescue lines from our previous work⁴. We have made some changes to the model and made it more specific, see the revised Figure 8b.

2. Does 5-HT treatment rescue the defect in FLP-7 secretion of *tph-1* mutants?

It does; we have included this in Figure 3.

3. Fig 6.e showed *npr-22* blocked *atgl-1* expression, what effect does *flp-7;npr-22* double mutant have on *atgl-1* expression?

We generated *flp-7;npr-22;atgl-1::GFP* mutants and found that they suppress the 5-HT-induced increase in *atgl-1* expression to the same extent as the *npr-22* mutants. The data is presented in Figure 7d.

4. Fig 3.d needs a legend to show what black and grey bars mean.

Amended.

5. Page 13: sentence fragment: "GR-159897 pre-treatment did not lead to at the doses tested."

Removed.

6. Could the authors comment on how FLP-7 signaling may function with *flp-18* (deBono, 2009) signaling to regulate fat content?

We have included a section in the discussion.

Reviewer 3

Here, Palamiuc, et al. identify a ligand-receptor pair that mediates the serotonergic control of fat metabolism, and identify the sites of ligand expression (ASI) and receptor activity (intestine). Overall, I think that this is a strong paper, so I will limit my comments to the most important points:

1. What regulates *flp-7* expression in ASI?

2. What role, if any, does CRTC-1 play in this pathway?

We have addressed points 1 and 2 together since they are experimentally linked. We discovered a role for the nutrient sensor AMPK in regulating FLP-7 release from the ASI neurons, thus defining a novel mechanism for neuropeptide secretion. We had anecdotally noted that *aak-2* mutants (the neuronal α subunit of AMPK) had decreased body fat to a similar extent as 5-HT treatment. Another study had suggested that AAK-2/AMPK mimicked the effects of increased 5-HT with respect to feeding behavior¹³. As described in the text accompanying Figure 6, further experimentation revealed that the decreased body fat phenotype of the *aak-2* null could be restored by re-introducing *aak-2* cDNA into the ASI neurons alone (Figure 6d). This suggested a role for AMPK in ASI neurons in regulating body fat. Prompted by this result, we next found that *aak-2* mutants had increased FLP-7 secretion, which was suppressed in the *aak-2;crtc-1* double mutant. Restoring *crtc-1* cDNA in ASI neurons in the double mutants subsequently 'uncovered' the increased secretion of FLP-7 in the *aak-2* mutant (Figure 6a,b). All the data supporting these results are now presented in Figure 6 with an accompanying Results section entitled 'AMPK signaling in the ASI neurons regulates FLP-7 secretion'. Taken together, we show that in the ASI neurons, the nutrient sensor AMPK regulates FLP-7 release via negative regulation of CRTC-1. Interestingly, because overexpression of AAK-2/AMPK increases lifespan via CRTC-1 regulation^{14, 15}, our proposed mechanism also suggests that mitigating high mitochondrial beta oxidation and rapid fat loss may underlie the previously noted increase in lifespan. We have mentioned this in the Discussion.

Minor Comments:

1. *Figure 1A seems out of place, since it summarizes published work rather than the work done here.*

Now removed.

2. *Was the 2008 screen mentioned done using Oil Red O or Nile Red? If the latter, do the results stand?*

In the 2008 screen, we had originally conducted the experiments using Nile Red. We verified all relevant mutants and 'hits' using Sudan Black B and lipid extraction followed by thin layer chromatography and mass spectrometry-based quantitation, also published in the same work. In a subsequent paper⁴, we again verified the key mutants using Oil Red O and biochemical extraction of triglycerides. The results stand with one exception: *tph-1* mutants have increased fat stores as judged by biochemical lipid extraction and Oil Red O⁴; this phenotype is not visible with vital Nile Red in our hands. Since we have never published on the *tph-1* mutant phenotype with Nile Red, this does not pose a problem. For more detail, please see Reviewer 2, Point 3.

3. *Were all of the flip/nlp/ins neuropeptides tested, or only a fraction? Important to state, even if flp-7 turned out to account for all of the activity (in case of redundancy).*

We screened all members of the neuropeptide and insulin gene families for which mutants exist (77/113). This point is mentioned in the Results section, and examples are now given in Figure 1b. Also see Reviewer 1, Point 1.

Literature Cited

1. Peckol EL, Troemel ER, Bargmann CI. Sensory experience and sensory activity regulate chemosensory receptor gene expression in *Caenorhabditis elegans*. *Proc Natl Acad Sci U S A* **98**, 11032-11038 (2001).
2. Meisel JD, Panda O, Mahanti P, Schroeder FC, Kim DH. Chemosensation of bacterial secondary metabolites modulates neuroendocrine signaling and behavior of *C. elegans*. *Cell* **159**, 267-280 (2014).
3. Srinivasan S, Sadegh L, Elle IC, Christensen AG, Faergeman NJ, Ashrafi K. Serotonin regulates *C. elegans* fat and feeding through independent molecular mechanisms. *Cell Metab* **7**, 533-544 (2008).
4. Noble T, Stieglitz J, Srinivasan S. An integrated serotonin and octopamine neuronal circuit directs the release of an endocrine signal to control *C. elegans* body fat. *Cell Metab* **18**, 672-684 (2013).
5. Witham E, Comunian C, Ratanpal H, Skora S, Zimmer M, Srinivasan S. *C. elegans* Body Cavity Neurons Are Homeostatic Sensors that Integrate Fluctuations in Oxygen Availability and Internal Nutrient Reserves. *Cell Rep* **14**, 1641-1654 (2016).
6. Elle IC, Olsen LC, Pultz D, Rodkaer SV, Faergeman NJ. Something worth dyeing for: molecular tools for the dissection of lipid metabolism in *Caenorhabditis elegans*. *FEBS Lett* **584**, 2183-2193 (2010).
7. Klapper M, *et al.* Fluorescence-based fixative and vital staining of lipid droplets in *Caenorhabditis elegans* reveal fat stores using microscopy and flow cytometry approaches. *J Lipid Res* **52**, 1281-1293 (2011).
8. Narasimhan SD, Yen K, Bansal A, Kwon ES, Padmanabhan S, Tissenbaum HA. PDP-1 links the TGF-beta and IIS pathways to regulate longevity, development, and metabolism. *PLoS Genet* **7**, e1001377 (2011).
9. O'Rourke EJ, Soukas AA, Carr CE, Ruvkun G. *C. elegans* major fats are stored in vesicles distinct from lysosome-related organelles. *Cell Metab* **10**, 430-435 (2009).
10. Soukas AA, Kane EA, Carr CE, Melo JA, Ruvkun G. Rictor/TORC2 regulates fat metabolism, feeding, growth, and life span in *Caenorhabditis elegans*. *Genes Dev* **23**, 496-511 (2009).
11. Xie M, Roy R. Increased levels of hydrogen peroxide induce a HIF-1-dependent modification of lipid metabolism in AMPK compromised *C. elegans* dauer larvae. *Cell Metab* **16**, 322-335 (2012).
12. Yen K, Le TT, Bansal A, Narasimhan SD, Cheng JX, Tissenbaum HA. A comparative study of fat storage quantitation in nematode *Caenorhabditis elegans* using label and label-free methods. *PLoS One* **5**, (2010).
13. Cunningham KA, *et al.* Loss of a neural AMP-activated kinase mimics the effects of elevated serotonin on fat, movement, and hormonal secretions. *PLoS Genet* **10**, e1004394 (2014).
14. Mair W, *et al.* Lifespan extension induced by AMPK and calcineurin is mediated by CRT-1 and CREB. *Nature* **470**, 404-408 (2011).
15. Burkewitz K, *et al.* Neuronal CRT-1 governs systemic mitochondrial metabolism and lifespan via a catecholamine signal. *Cell* **160**, 842-855 (2015).

Reviewer #1 (Remarks to the Author)

The authors have done a great job in addressing the reviewers' comments. They provide an important additional insight establishing a link between AMPK/CRTC and FLP-7/NPR-22 signaling in ASI neurons. I think the manuscript has greatly improved and recommend it for publication in Nature Communications.

Reviewer #2 (Remarks to the Author)

As reviewers pointed out, several prior papers have identified neuron-to-intestine signaling regulating fat content in *C. elegans*, including the role of serotonin in regulating fat (deBonno lab 2009, Ashrafi lab 2014 and the authors prior work). Neuronal *aak-2* has been previously implicated in fat control and in neurosecretion (Ashrafi lab, 2014), and *crtc-1* has been shown to function downstream of *aat-2* (Dillon lab 2011 and others), so outside of the idea that *flp-7* secretion is regulated (which is not convincing, see below), the new ground covered in this study is somewhat limited.

Major concerns:

The authors did not adequately address whether the changes in coelomocyte fluorescence observed in Figs 3, 4, 5 and 6 reflect changes in the regulated secretion of FLP-7 (as the authors conclude) or whether they reflect changes in the expression of the *daf-7* promoter used to express the FLP-7-cherry transgenes in ASI. This is a critical point because the idea that FLP-7 secretion is regulated by fat signaling pathways in the nervous system is the major novel argument made in this study, but this conclusion is still not supported by the data.

The problem hinges on the use of the *daf-7* promoter to drive expression of *flp-7* cherry. This promoter was selected because it is expressed in ASI, where *flp-7* functions, but expression from this promoter is regulated by several environmental inputs, including by serotonin signaling circuits. For example, Ruvkun lab, 2000, and this study (fig S3b) show that transcription from the *daf-7* promoter is reduced in *tph-1* mutants. Thus the decrease in *flp-7*-cherry coelomocyte fluorescence observed in *tph-1* mutants (Fig b-d) likely reflects a decrease in the expression of *flp-7* from the *daf-7* promoter rather than a decrease in the regulation of secretion, as the authors suggest (see model in Fig 8). Therefore, the *tph-1* data in Fig 3b-d and Fig5b-d is uninterpretable and should be removed.

On a positive note, the authors added a control that shows that there is no increase in the expression of *daf-7::GFP* in animals exposed to 5HT (Fig S3c), supporting the idea that exogenous 5HT promotes neuro-secretion. This was the only condition for which this control was done (Why??), and thus is the only FLP-7 secretion result that was properly interpreted. However, this result is of limited novelty because it was already shown (Ashrafi, 2014) and this does not address whether endogenous serotonin regulates FLP-7 secretion.

More concerning is that the authors did not conduct the appropriate controls to determine if any of the other mutants used in this study (including *mod-5*, *mod-1*, *ser-6*, *aak-2*, *crtc-1*) alter *daf-7* expression. Instead of analyzing *daf-7::GFP* expression in ASI by quantitative fluorescence microscopy, which is the gold standard control, they utilized qRT-PCR (in some, not all of these mutants). The major problem with using qRT-PCR for this purpose is that *daf-7* is only expressed one cell, making it difficult to reliably quantify any small changes in expression. The difficulty with qRT-PCR is illustrated by the authors own data: First, the authors were only able to detect a 50% reduction in *daf-7* expression in *tph-1* mutants (Fig S3b) compared to a 6.5 fold reduction reported by Ruvkun et al, who used the *daf-7::GFP* transcriptional reporter. Second, examination of *daf-7* expression in *tph-1* mutants gave conflicting results (FigS3a and b). After demonstrating that *tph-1* mutants regulate *daf-7* expression, the authors should not have kept any of the coelomocyte data using *tph-1* mutants in the paper. In Figs 3 and 4, the inability of 5HT application to restore normal coelomocyte fluorescence to *tph-1* or *ser-6* mutants is inconsistent with the model presented in Fig 8 that these genes function upstream of FLP-7 to promote its secretion, but instead further confirms the idea that these mutants alter expression of the *daf-7*-promoted *flp-7* transgene.

Finally, in response to reviewers suggestions to utilize a promoter that is not regulated by serotonin, the authors selected the str-3 promoter and referenced two papers that they say show that this promoter is not regulated by tph-1 or by 5HT. I could find no experiments in those two papers supporting this claim, and the authors performed no controls to demonstrate that this promoter is not regulated by serotonin signaling mutants in their hands. Also, they only examined two conditions (5HT and mod-5) using this promoter citing inability to quantify changes due to low expression of this transgene. This technical problem could have been circumvented by generating additional transgenic lines or by utilizing one of the other ASI-specific promoters, neither of which the authors did.

Conclusions that are not supported by the data:

Line 241: "Together, our results suggest that mod-1 and ser-6 are both required for the secretion of FLP-7." is not supported by the data since these mutants have no effect on flp-7 secretion (Fig 4).

Line 216 "tph-1 mutants showed a significant decrease in FLP-7 secretion (~60% of wildtype)...(Fig. 3b-d)", and line 221 "these experiments show that FLP-7 secretion from the ASI neurons is responsive to changes in neuronal 5-HT signaling, and that it drives fat loss in the intestine." are not justified because tph-1 mutants reduce transgene expression (fig S3b).

Line 254. "as predicted, tph-1 mutants also showed a decrease in OA-induced FLP-7mCherry secretion (Fig. 5b-d)." The reduction could be explained by a reduction of expression of the daf-7 promoter in tph-1 mutants (Ruvkun et al and Fig S3b), so to say that FLP-7mCherry secretion is regulated by tph-1 is not justified.

Other concerns:

1) While the identification of FLP-7 as a signal that contributes to fat regulation by exogenous serotonin is a novel finding, the effects of flp-7 mutations are much more modest than those of unc-31/CAPS mutants (compare Fig 1a and 1c). Whereas unc-31 mutants have increased fat content, flp-7 mutants are similar to wild type (Fig 1a and 1c). Thus other neuroendocrine signals are likely to have much more prominent roles in fat metabolism than flp-7. The question of whether the modest effects of flp-7 mutations are associated with physiologically relevant consequence on lifespan or health would be of interest, but was not explored.

2) The authors state in the Abstract: "This ligand-receptor pair does not alter other serotonin-dependent behaviors...". However, in Fig1h, flp-7 mutants seem to have a strong egg-laying defect (lay less than half the number of eggs compared to wild type). It is puzzling that the authors do not show whether this difference is statistically significant (based on the error bars, it should be) or mention the existence of the defect in the text. This data is important for two reasons: first, it suggests that flp-7 may not be specifically required for fat regulation, as the authors conclude. Second, it brings up the question of whether the (rather modest) defects in fat metabolism of flp-7 mutants may be a secondary effect of their defects in egg laying, a central question that the authors did not address.

3) Authors were asked to re-evaluate the sequence homology between flp-7 and substance P using the mature (not immature) peptides. However, Fig S1 still shows the sequence comparison with the immature peptides, and authors state in line 138 "The flp-7 peptides that resemble the mammalian tachykinin peptide family exemplified by Substance P." Substance P does not come up in searches using flp7 and vice versa. Thus any sequence 'resemblance' is not likely to indicate evolutionary conservation.

4) The authors still miss-state that npr-22 is a tachykinin receptor 2 ortholog. Tkr-1, 2 and 3 also share as much or more sequence similarity to tachykinin receptor 2 as npr-22 does. Indeed, npr-22 is not the best reciprocal hit to TKR2 (tkr-2 comes in first at e-40). It would be more accurate to refer to npr-22 as a tachykinin receptor 2 homolog.

Reviewer #3 (Remarks to the Author)

The authors have addressed all my concerns, and I appreciate the addition of new experiments--I think that their model nicely incorporates all of the data now.

2nd point-by-point response to Reviewers' comments

Three reviewers evaluated our initial submission, and overall found our work rigorous and interesting. They each asked for several additional experiments to strengthen the claims of the manuscript, including measuring transcript levels of the neuropeptide under study, (Reviewers 1 and 2), additional transgenic lines using new promoters (Reviewer 2), and the mechanism of action in the neuropeptide-secreting ASI neurons (Reviewer 3). In our resubmission, we addressed *all* experimental concerns with extensive new data, including identification of a novel mechanism of neuropeptide release based on AMPK/CTRC-1 signaling. We believe that having responded with new experimental data to the initial set of comments strengthened the manuscript considerably.

Reviewers 1 and 3 were satisfied with the revised manuscript (see comments). Reviewer 2, on the other hand, has raised new concerns not originally stated, and re-stated concerns which cannot be justified given the presented data. It is my opinion that the new comments and experiments from Reviewer 2 are either unreasonable, or are not essential to support the conclusions that we draw. We have addressed each of the points under Reviewer 2, below.

Reviewers' comments:

Reviewer #1 (Remarks to the Author):

The authors have done in great job in addressing the reviewers' comments. They provide an important additional insight establishing a link between AMPK/CRTC and FLP-7/NPR-22 signaling in ASI neurons. I think the manuscript has greatly improved and recommend it for publication in Nature Communications.

We thank Reviewer #1 for their comments, and their help in strengthening the work.

Reviewer #2 (Remarks to the Author):

*As reviewers pointed out, several prior papers have identified neuron-to-intestine signaling regulating fat content in *C. elegans*, including the role of serotonin in regulating fat (deBonno lab 2009, Ashrafi lab 2014 and the authors prior work). Neuronal *aak-2* has been previously implicated in fat control and in neurosecretion (Ashrafi lab, 2014), and *crtc-1* has been shown to function downstream of *aat-2* (Dillon lab 2011 and others), so outside of the idea that *flp-7* secretion is regulated (which is not convincing, see below), the new ground covered in this study is somewhat limited.*

No study to date has identified a secreted neuroendocrine factor that is responsive to neuronal serotonin signaling, identified its cognate receptor and site of action, and demonstrated the mechanism of fat regulation. Our previous work¹ defined the neural circuit for serotonin, and although the work suggested that there ought to exist novel neuroendocrine factors, they were not identified in that study. The deBono 2009 study did identify neuropeptides and receptors related to vital Nile Red staining². However, vital Nile Red staining has been called into question through the work of several laboratories over the last few years, and is now widely recognized as an incomplete measure of body fat content³⁻⁸. Although an interesting body of work, the deBono 2009 study did not demonstrate that the identified neuropeptide (*flp-18*) is secreted and thus functions as bona fide neuroendocrine factor. Additionally, *flp-18* has several roles in coordinating behavior and Nile Red granules, and it is still unclear whether *flp-18* truly regulates lipid metabolism. The Ashrafi 2014 work does suggest that *aak-2* mutants have increased

neurosecretion⁹. However, the readout used (daf-28 peptide secretion) is artificial in the sense that it is neither a consequence of serotonergic signaling, nor instructive for physiological outputs of neuronal serotonin. Rather, it serves as a useful readout of secretion in response to loss of *aak-2*. Thus, we believe that our report is significant in the following way: it is the first identification of a secreted hormone or neuroendocrine factor that functions in response to neuronal serotonergic signaling, and functions to stimulate fat metabolism in a distal tissue. We also identify a functional role for its cognate receptor in vivo. A role for AAK-2 (neuronal AMPK α -subunit) opens the door to future investigations of the mechanisms by which the ASI neurons integrate nutrient-sensing information, with coordinating neuropeptide release. We show that FLP-7 signaling selectively stimulates serotonin-mediated fat loss and not other outputs of neuronal serotonin such as feeding behavior, locomotion or reproduction. Our work uncovers a novel conceptual framework in which global modulators such as serotonin can modulate physiological outputs independently of one another, by utilizing distinct neuropeptides/endocrine factors. Thus, we respectfully disagree that the new ground covered is somewhat limited, and believe the findings and implications of this study go well beyond the above-mentioned previous studies.

Major concerns:

1. *The authors did not adequately address whether the changes in coelomocyte fluorescence observed in Figs 3, 4, 5 and 6 reflect changes in the regulated secretion of FLP-7 (as the authors conclude) or whether they reflect changes in the expression of the daf-7 promoter used to express the FLP-7-cherry transgenes in ASI. This is a critical point because the idea that FLP-7 secretion is regulated by fat signaling pathways in the nervous system is the major novel argument made in this study, but this conclusion is still not supported by the data.*

The problem hinges on the use of the daf-7 promoter to drive expression of flp-7 cherry. This promoter was selected because it is expressed in ASI, where flp-7 functions, but expression from this promoter is regulated by several environmental inputs, including by serotonin signaling circuits. For example, Ruvkun lab, 2000, and this study (fig S3b) show that transcription from the daf-7 promoter is reduced in tph-1 mutants. Thus the decrease in flp-7-cherry coelomocyte fluorescence observed in tph-1 mutants (Fig b-d) likely reflects a decrease in the expression of flp-7 from the daf-7 promoter rather than a decrease in the regulation of secretion, as the authors suggest (see model in Fig 8). Therefore, the tph-1 data in Fig 3b-d and Fig5b-d is uninterpretable and should be removed.

Based on the original concerns raised by Reviewer #2, in which the reported effects of *tph-1* mutants on the *daf-7* promoter were mentioned, we measured the mRNA of the *flp-7mCherry* transgene as well as endogenous *daf-7* mRNA in *tph-1* and *mod-5* mutants, and with exogenous 5-HT treatment (Supplementary Figures 3a and b). In these experiments we found that, relative to wild-type controls, the *flp-7mCherry* transcript levels were unaltered in the genotypes/conditions tested. On the other hand, as previous reports indicate, *daf-7* expression is indeed decreased in *tph-1* mutants, but not in *mod-5* or in 5-HT treatment.

There are two important differences to consider when comparing the mRNA levels of *daf-7* and *flp-7mCherry*, both of which we had stated in our original point-by-point response. First, the *flp-7mCherry* transgenic reporter is overexpressed, integrated and outcrossed and therefore may not be regulated as would endogenous *daf-7*. Second is the possibility of promoter-independent effects via 3'-untranslated regions or 3'UTRs, which in *C. elegans* significantly regulate mRNA expression in addition to effects on translation (www.wormbook.org). The 3'UTR is different for the *flp-7mCherry* transgene and for the endogenous *daf-7* gene. For the *flp-7mCherry* reporter, we used the *unc-54* 3'UTR, which is

widely chosen in the field because of its ability to confer robust transgene expression. The endogenous *daf-7* gene would have its own 3'UTR, and would therefore be expected to be regulated differently from the exogenous transgene. Either or both of these reasons might explain why *tph-1* mutants regulate endogenous *daf-7* mRNA but not the *flp-7mCherry* transgene mRNA. Regardless, because *flp-7mCherry* mRNA levels are not regulated by *tph-1*, *mod-5* or exogenous 5-HT (Supplementary Figure 3a), we believe that changes in secretion of the FLP-7mCherry fusion protein in these mutants have been correctly interpreted, and the observed changes are a function of altered neuronal secretion, and not mRNA levels. These conclusions are also supported by having obtained similar results from an independent transgenic line using a different ASI-specific promoter, *str-3* (see point 6 below; Supplementary Figures 3 d, e).

2. On a positive note, the authors added a control that shows that there is no increase in the expression of daf-7::GFP in animals exposed to 5HT (Fig S3c), supporting the idea that exogenous 5HT promotes neuro-secretion. This was the only condition for which this control was done (Why??), and thus is the only FLP-7 secretion result that was properly interpreted. However, this result is of limited novelty because it was already shown (Ashrafi, 2014) and this does not address whether endogenous serotonin regulates FLP-7 secretion.

Because our qPCR results had already shown that *flp-7mCherry* transgene mRNA levels were not altered by the serotonin signaling mutants, crossing the *daf-7::GFP* reporter (which we believe has a much lower dynamic range than qPCR, please see point 4 below) would yield little further insight into *flp-7mCherry* regulation. *flp-7mCherry* mRNA was not altered in the endogenous signaling mutants (*tph-1* and *mod-5*; Supplementary Figure 3a), therefore our data do support the conclusion that endogenous serotonin regulates FLP-7mCherry peptide secretion. We also do not believe that this result is of limited novelty because FLP-7 has never previously identified as functioning in the serotonergic pathway, including in the Ashrafi 2014 study⁹.

3. More concerning is that the authors did not conduct the appropriate controls to determine if any of the other mutants used in this study (including mod-5, mod-1, ser-6, aak-2, crtc-1) alter daf-7 expression.

Please see point #2 above. We did examine *mod-5* mutants (Supplementary Figure 3a, b). We did not examine the other mutants because the dominant 'upstream' controllers of endogenous 5-HT (*tph-1* and *mod-5*) as well as exogenous 5-HT did not alter *flp-7mCherry* mRNA. We reiterate that the relevant control for measuring secretion of the FLP-7mCherry protein is the *flp-7mCherry* mRNA, not *daf-7* mRNA as explained in points 1 and 2.

4. Instead of analyzing daf-7::GFP expression in ASI by quantitative fluorescence microscopy, which is the gold standard control, they utilized qRT-PCR (in some, not all of these mutants). The major problem with using qRT-PCR for this purpose is that daf-7 is only expressed one cell, making it difficult to reliably quantify any small changes in expression. The difficulty with qRT-PCR is illustrated by the authors own data: First, the authors were only able to detect a 50% reduction in daf-7 expression in tph-1 mutants (Fig S3b) compared to a 6.5 fold reduction reported by Ruvkun et al, who used the daf-7::GFP transcriptional reporter. Second, examination of daf-7 expression in tph-1 mutants gave conflicting results (FigS3a and b). After demonstrating that tph-1 mutants regulate daf-7 expression, the authors should not have kept any of the coelomocyte data using tph-1 mutants in the paper.

We respectfully differ in our view of the gold standard control. Measuring fluorescence intensity in the *daf-7::GFP* reporter line entails recording GFP levels as driven by the *daf-7* promoter. Thus, we would be measuring steady state levels of the GFP protein, not mRNA. We and many other labs have used qPCR on non-abundant transcripts to record quantitative differences in mRNA in *C. elegans*¹⁰⁻¹⁵, and in many ways, this method is recognized as having a greater dynamic range and being more precise for mRNA expression than steady state GFP levels. We cannot comment on the 6.5-fold change mentioned above in the Ruvkun et al. study. However, in the ~12-15 GFP-based reporters we have studied closely in my lab, we have never encountered fold changes of such magnitude by measuring GFP using fluorescence microscopy. As stated in points 1 and 2 above, because *tph-1* mutants do not show changes in *flp-7mCherry* mRNA, which is the relevant control (rather than *daf-7* mRNA), our results do support the conclusion that *tph-1* mutants have decreased FLP-7mCherry secretion.

5. In Figs 3 and 4, the inability of 5HT application to restore normal coelomocyte fluorescence to tph-1 or ser-6 mutants is inconsistent with the model presented in Fig 8 that these genes function upstream of FLP-7 to promote its secretion, but instead further confirms the idea that these mutants alter expression of the daf-7-promoted flp-7 transgene.

We disagree. In Figure 3 panels b-d, we show that 5-HT administration does in fact significantly ($p < 0.05$) restore FLP-7mCherry secretion to *tph-1* mutants, and is comparable to the increase elicited in wild-type animals. This is because *tph-1* mutants lack 5-HT, and its exogenous addition restores the defect in FLP-7 secretion. On the other hand, *ser-6* mutants block 5-HT-mediated FLP-7mCherry secretion (Figure 4) as they do 5-HT-mediated fat loss, as previously described in our Noble et al 2013 study. In this sense, the data presented in Figure 4 do fit well with the model presented in Figure 8. Additionally. Please see point 1-4 above. Regulatory mutants of the 5-HT pathway including *tph-1* do not alter *flp-7mCherry* mRNA (Supplementary Figure 3a), which is the relevant control for FLP-7mCherry protein secretion.

6. Finally, in response to reviewers suggestions to utilize a promoter that is not regulated by serotonin, the authors selected the str-3 promoter and referenced two papers that they say show that this promoter is not regulated by tph-1 or by 5HT. I could find no experiments in those two papers supporting this claim, and the authors preformed no controls to demonstrate that this promoter is not regulated by serotonin signaling mutants in their hands. Also, they only examined two conditions (5HT and mod-5) using this promoter citing inability to quantify changes due to low expression of this transgene. This technical problem could have been circumvented by generating additional transgeneic lines or by utilizing one of the other ASI-specific promoters, neither of which the authors did.

Prompted by the original comments of the reviewer, we generated new transgenic lines with a different ASI-specific promoter, to evaluate secretion of FLP-7mCherry by independent means. We chose *str-3* because it is the most widely-used ASI promoter other than *daf-7*¹⁶⁻¹⁸, and is not known to be regulated by serotonin. In this sense, it represents the best choice for ASI control (other than *daf-7*). It is also known to be a weaker promoter than *daf-7*. We did in fact generate several transgenic lines for the *str-3::flp-7mCherry* construct. For all our *str-3* lines, we obtained low transgene expression relative to the *daf-7* promoter, which we had expected. Despite the decreased baseline levels, the quantitative increase in FLP-7mCherry secretion with endogenous (*mod-5*) and exogenous 5-HT increases were comparable to the results with the *daf-7* promoter (please see Figure 3d and Supplementary Figure 3e). We could have crossed the transgenic line into the *tph-1* mutant. However, it was clear that given the rather low

baseline expression in wild-type animals (Supplementary Figure 3d), demonstrating a further reduction with *tph-1* would not have been a logical or interpretable experiment. Furthermore, we did evaluate another ASI-specific promoter, *gpa-4*, for which we also observed an increase in FLP-7mCherry secretion with 5-HT treatment (unpublished observations). However, we used this promoter to control expression of *aak-2* and *crtc-1* to the ASI neurons (Figure 6) because we needed another independent promoter to control these genes in the *flp-7mCherry* secretion lines. To our knowledge there are no other ASI-specific promoters.

Conclusions that are not supported by the data:

7. Line 241: *“Together, our results suggest that mod-1 and ser-6 are both required for the secretion of FLP-7.” is not supported by the data since these mutants have no effect on flp-7 secretion (Fig 4).*

Although the mutants themselves are dispensable for FLP-7 secretion, they are required for the 5-HT-dependent increase in secretion. We have amended the text to make this point clear.

8. Line 216 *“tph-1 mutants showed a significant decrease in FLP-7 secretion (~60% of wildtype)... (Fig. 3b-d)”*, and line 221 *“these experiments show that FLP-7 secretion from the ASI neurons is responsive to changes in neuronal 5-HT signaling, and that it drives fat loss in the intestine.”* are not justified because *tph-1* mutants reduce transgene expression (fig S3b). Line 254. *“as predicted, tph-1 mutants also showed a decrease in OA-induced FLP-7mCherry secretion (Fig. 5b-d).”* The reduction could be explained by a reduction of expression of the *daf-7* promoter in *tph-1* mutants (Ruvkun et al and Fig S3b), so to say that *FLP-7mCherry* secretion is regulated by *tph-1* is not justified.

Please see points 1-5 above.

Other concerns:

9. *While the identification of FLP-7 as a signal that contributes to fat regulation by exogenous serotonin is a novel finding, the effects of flp-7 mutations are much more modest than those of unc-31/CAPS mutants (compare Fig 1a and 1c). Whereas unc-31 mutants have increased fat content, flp-7 mutants are similar to wild type (Fig 1a and 1c). Thus other neuroendocrine signals are likely to have much more prominent roles in fat metabolism than flp-7. The question of whether the modest effects of flp-7 mutations are associated with physiologically relevant consequence on lifespan or health would be of interest, but was not explored.*

unc-31 mutants lack pan-neuronal secretion of neuropeptides and biogenic amines including 5-HT and octopamine. We do not doubt that there are other neuropeptides, unrelated to the serotonergic pathway, that are required for fat metabolism and that are not secreted in the *unc-31* mutants. Thus, although we agree that other neuropeptides are likely to have important roles in fat metabolism, it is an open question as to whether they will (i) turn out to be true neuroendocrine molecules and (ii) be more or less prominent than *flp-7*. The differences between the *unc-31* and *flp-7* fat phenotypes do not indicate a hierarchical relationship of relative prominence. Rather, these differences highlight the importance of undertaking studies such as this, to identify additional neuronal regulators of peripheral metabolism, that require *unc-31* for their secretion. In fact, *flp-7* mutants do not have a fat phenotype on their own (Figures 1b, c). Rather, our data show that *flp-7* is recruited in response to serotonin signaling to mediate the effects on fat loss in the intestine, and is an instructive regulator of fat metabolism

because its over-expression drives fat loss in an ATGL-1-dependent manner (Figure 1d).

With respect to a role in lifespan: we have not observed a clear change in lifespan with neuronal serotonin mutants (our unpublished observations), and thus did not see the rationale for studying *flp-7* mutants in this context. We also consider the study of lifespan to be outside the scope of this work.

10. The authors state in the Abstract: "This ligand-receptor pair does not alter other serotonin-dependent behaviors...". However, in Fig1h, flp-7 mutants seem to have a strong egg-laying defect (lay less than half the number of eggs compared to wild type). It is puzzling that the authors do not show whether this difference is statistically significant (based on the error bars, it should be) or mention the existence of the defect in the text. This data is important for two reasons: first, it suggests that flp-7 may not be specifically required for fat regulation, as the authors conclude. Second, it brings up the question of whether the (rather modest) defects in fat metabolism of flp-7 mutants may be a secondary effect of their defects in egg laying, a central question that the authors did not address.

As the reviewer points out, untreated *flp-7* mutants do have a defect in egg-laying, and we have now amended the text to state this explicitly, and modified the graph in Figure 1h to include significance. However, this result does not refute the conclusion that *flp-7* is required specifically for 5-HT-mediated fat regulation. First, the claim of the manuscript is not that the *flp-7* mutants have defects in fat metabolism. In fact, *flp-7* mutants (like *mod-1* mutants, Noble et al 2013) do not have any change in body fat (Figure 1b, c; see point 9 above). Rather, the manuscript claims that *flp-7* mutants suppress 5-HT-mediated fat loss without suppressing 5-HT-mediated changes in food intake, egg-laying and locomotion (Figure 1c-i). In other words, *flp-7* is dispensable for fat metabolism in an otherwise wild-type background. However, an increase in neuronal 5-HT signaling requires the secretion of FLP-7 to elicit the effects on fat metabolism in the intestine, without which 5-HT induced fat loss does not occur.

Although the *flp-7* mutants have an egg-laying defect on their own, they do not have a fat phenotype. Thus, our results support the opposite of the stated concern: that defects in egg-laying do not predispose or predict a shift in fat metabolism. *flp-7* mutants do suppress 5-HT-mediated fat loss (Figure 1c), but do not suppress 5-HT-mediated increase in egg-laying (Figure 1h), thus neither fat metabolism nor egg-laying are predictive or consequential to one another in this context. In a previous study¹⁴ we had shown that the serotonergic pathway for egg-laying is distinct from that of fat metabolism by examining several genes in the serotonergic egg-laying pathway for changes in fat content. Additionally, MOD-1, the critical fat-regulatory serotonergic receptor is not required for 5-HT-dependent egg-laying¹⁹. Together, our results do not support the conclusion that the fat metabolism of *flp-7* mutants are a secondary effect of defects in egg-laying but rather, that they are independent of one another: *flp-7* mutants have an egg-laying defect but no change in fat metabolism, and *flp-7* mutants suppress 5-HT-mediated fat loss but not 5-HT-mediated egg-laying.

FLP-7 is expressed in a few neurons besides ASI (Figure 2a), and the egg-laying defect of *flp-7* mutants may be a consequence of its role in other neurons, unrelated to the serotonergic circuit. We have made this point in the discussion.

11. Authors were asked to re-evaluate the sequence homology between flp-7 and substance P using the mature (not immature) peptides. However, Fig S1 still shows the sequence

comparison with the immature peptides, and authors state in line 138 "The flp-7 peptides that resemble the mammalian tachykinin peptide family exemplified by Substance P." Substance P does not come up in searches using flp7 and vice versa. Thus any sequence 'resemblance' is not likely to indicate evolutionary conservation.

We agree that the FLP-7- Substance P sequence resemblance may or may not predict evolutionary conservation, and given the short sequence, would not expect it to be revealed in searches.

12. The authors still miss-state that npr-22 is a tachykinin receptor 2 ortholog. Tkr-1, 2 and 3 also share as much or more sequence similarity to tachykinin receptor 2 as npr-22 does. Indeed, npr-22 is not the best reciprocal hit to TKR2 (tkr-2 comes in first at e-40). It would be more accurate to refer to npr-22 as a tachykinin receptor 2 homolog.

We have amended the text to state "npr-22 is a *C. elegans* ortholog" rather than "the ortholog". "Homolog" could also refer to genes within the same species and could lead to confusion.

Reviewer #3 (Remarks to the Author):

The authors have addressed all my concerns, and I appreciate the addition of new experiments- I think that their model nicely incorporates all of the data now.

We thank Reviewer #3 for their comments and support of the work.

Literature Cited

- 1 Noble, T. *et al.* An integrated serotonin and octopamine neuronal circuit directs the release of an endocrine signal to control *C. elegans* body fat *Cell Metab* **18**, 672-684, (2013).
- 2 Cohen, M. *et al.* Coordinated regulation of foraging and metabolism in *C. elegans* by RFamide neuropeptide signaling *Cell Metab* **9**, 375-385, (2009).
- 3 Yen, K. *et al.* A comparative study of fat storage quantitation in nematode *Caenorhabditis elegans* using label and label-free methods *PLoS One* **5**, (2010).
- 4 Klapper, M. *et al.* Fluorescence-based fixative and vital staining of lipid droplets in *Caenorhabditis elegans* reveal fat stores using microscopy and flow cytometry approaches *J Lipid Res* **52**, 1281-1293, (2011).
- 5 Narasimhan, S. D. *et al.* PDP-1 links the TGF-beta and IIS pathways to regulate longevity, development, and metabolism *PLoS Genet* **7**, e1001377, (2011).
- 6 O'Rourke, E. J. *et al.* *C. elegans* major fats are stored in vesicles distinct from lysosome-related organelles *Cell Metab* **10**, 430-435, (2009).
- 7 Soukas, A. A. *et al.* Rictor/TORC2 regulates fat metabolism, feeding, growth, and life span in *Caenorhabditis elegans* *Genes & Development* **23** 496-511 (2009).
- 8 Xie, M. & Roy, R. Increased levels of hydrogen peroxide induce a HIF-1-dependent modification of lipid metabolism in AMPK compromised *C. elegans* dauer larvae *Cell Metab* **16**, 322-335, (2012).
- 9 Cunningham, K. A. *et al.* Loss of a neural AMP-activated kinase mimics the effects of elevated serotonin on fat, movement, and hormonal secretions *PLoS Genet* **10**, e1004394, (2014).

- 10 Ratnappan, R. *et al.* Germline signals deploy NHR-49 to modulate fatty-acid beta-oxidation and desaturation in somatic tissues of *C. elegans* *PLoS Genet* **10**, e1004829, (2014).
- 11 Taubert, S. *et al.* A Mediator subunit, MDT-15, integrates regulation of fatty acid metabolism by NHR-49-dependent and -independent pathways in *C. elegans* *Genes Dev* **20**, 1137-1149, (2006).
- 12 Van Gilst, M. R. *et al.* Nuclear hormone receptor NHR-49 controls fat consumption and fatty acid composition in *C. elegans* *PLoS Biol* **3**, e53, (2005).
- 13 Van Gilst, M. R. *et al.* A *Caenorhabditis elegans* nutrient response system partially dependent on nuclear receptor NHR-49 *Proc Natl Acad Sci U S A* **102**, 13496-13501, (2005).
- 14 Srinivasan, S. *et al.* Serotonin regulates *C. elegans* fat and feeding through independent molecular mechanisms *Cell Metab* **7**, 533-544, (2008).
- 15 Walker, A. K. *et al.* A conserved SREBP-1/phosphatidylcholine feedback circuit regulates lipogenesis in metazoans *Cell* **147**, 840-852, (2011).
- 16 Meisel, J. D. *et al.* Chemosensation of bacterial secondary metabolites modulates neuroendocrine signaling and behavior of *C. elegans* *Cell* **159**, 267-280, (2014).
- 17 Peckol, E. L. *et al.* Sensory experience and sensory activity regulate chemosensory receptor gene expression in *Caenorhabditis elegans* *Proc Natl Acad Sci U S A* **98**, 11032-11038, (2001).
- 18 Calhoun, A. J. *et al.* Neural Mechanisms for Evaluating Environmental Variability in *Caenorhabditis elegans* *Neuron* **86**, 428-441, (2015).
- 19 Ranganathan, R. *et al.* MOD-1 is a serotonin-gated chloride channel that modulates locomotory behaviour in *C. elegans* *Nature* **408**, 470-475, (2000).

Reviewer #4 (Remarks to the Author)

I am reviewing this after seeing the other reviews.

The story is definitely interesting, and represents a significant advance, and thus should certainly not be scuttled. The key remaining issue is whether flp-:GFP levels in ASI change under the various conditions or if the effects of serotonin can primarily be ascribed to FLP-7 secretion. The whole-animal qPCR is tough to get appropriate controls for the ability to detect changes, so the quantitation of fluorescence should be done for a reasonable set of the strains.

If it is inconclusive (as it well might be), then you will have to back off your claims. Indeed, I suggest that you be more conservative throughout.

Minor points:

As I looked through your strain list to see what reagents you had in hand, I noticed that you are not defining genotypes according to standard nomenclature. For example, a transgene should have a unique name (such as nIs123 or stEx456).

Point-by-point response to Reviewer #4 comments

Reviewer #4 independently reviewed the manuscript and the previous point-by-point responses in response to Reviewers #1-3. We have answered the questions raised by this reviewer in the point-by-point response, below.

Reviewer's comments:

Reviewer #4 (Remarks to the Author):

I am reviewing this after seeing the other reviews.

The story is definitely interesting, and represents a significant advance, and thus should certainly not be scuttled. The key remaining issue is whether flp::GFP levels in ASI change under the various conditions or if the effects of serotonin can primarily be ascribed to FLP-7 secretion. The whole-animal qPCR is tough to get appropriate controls for the ability to detect changes, so the quantitation of fluorescence should be done for a reasonable set of the strains. If it is inconclusive (as it well might be), then you will have to back off your claims. Indeed, I suggest that you be more conservative throughout.

We thank the Reviewer for their comment regarding the significance of the manuscript. We have conducted the requested experiments and measured the fluorescence intensity of FLP-7 in the ASI neurons in all of the key serotonergic mutant strains, and did not observe any differences across all genotypes and conditions tested. The data are presented in Supplementary Figures 3d and e. On the basis of the fluorescence intensity measurements and the qPCR experiments, we conclude that the serotonergic pathway influences FLP-7 secretion.

The changes are highlighted in the text file.

Minor points:

As I looked through your strain list to see what reagents you had in hand, I noticed that you are not defining genotypes according to standard nomenclature. For example, a transgene should have a unique name (such as nls123 or stEx456).

We thank the Reviewer for this reminder. We have now correctly named the strains (Supplementary Table 1) per the standard *C. elegans* nomenclature, as defined on WormBase.

Reviewer #4 (Remarks to the Author)

I am now satisfied.